# PokéLLMon: A Grounding and Reasoning Benchmark for Large Language Models in Pokémon Battles

## Abstract

Developing grounding techniques for LLMs poses two requirements for interactive environments, *i.e.*, (i) the presence of rich knowledge beyond the scope of existing LLMs and (ii) the complexity of tasks that require strategic reasoning. Existing environments fail to meet both requirements due to their simplicity or reliance on commonsense knowledge already encoded in LLMs for interaction. In this paper, we present PokéLLMon, a new benchmark enriched with fictional game knowledge and characterized by the intense, dynamic, and adversarial gameplay of Pokémon battles, setting new challenges for the development of grounding and reasoning techniques in interactive environments. Empirical evaluations demonstrate that existing LLMs lack game knowledge and struggle in Pokémon battles. We investigate grounding techniques that leverage game knowledge and self-play experience, and provide a thorough analysis of reasoning methods from a new perspective of action consistency. Additionally, we introduce higher-level reasoning challenges when playing against human players. The implementation of our benchmark is anonymously released at: https://anonymous.4open.science/r/PokeLLMon.

## 1 Introduction

The success of Large Language Models (LLMs) comes from encoding massive textual data by predicting the next token with huge model capacity, and generalizing well to various tasks (Ouyang et al., 2022; Brown et al., 2020; Achiam et al., 2023; Xi et al., 2023; Wang et al., 2023b). The pre-trained LLMs, encode the knowledge from text and exhibit cognitive abilities such as reasoning and planning, ways of organizing knowledge described in text inherently.

In other words, LLMs lack experiential grounding (Mahowald et al., 2024; Hu et al., 2024), which prevents them from understanding new concepts outside their scope, or evolving their reasoning abilities in the interactive environments. Recent research focuses on grounding LLMs in games via Reinforcement Learning (RL) (Carta et al., 2023; Tan et al., 2024) or supervised fine-tuning (Zhu et al., 2023; Feng et al., 2023). However, it has been observed that even a well-trained agent still lacks generalizability to slightly altered settings, such as substituting action tokens with synonyms (Carta et al., 2023), suggesting that simplistic environments with limited scenarios do not best facilitate the development of grounding and reasoning techniques.

Developing grounding and reasoning methods for LLMs places two requirements for interactive environments: (i) the presence of knowledge beyond the scope of existing LLMs and (ii) the complexity of tasks that demand strategic reasoning abilities. The environments used in previous work do not meet both of these requirements due to their reliance on commonsense knowledge encoded in LLMs (Shridhar et al., 2020; Xiang et al., 2024) (a locked door can be unlocked with a key), or their simplicity and limited number of scenarios (Carta et al., 2023; Tan et al., 2024).

**Scope and Contributions:** In this paper, we introduce PokéLLMon, a new grounding and reasoning benchmark for LLMs in Pokémon Battles, which sets a new challenge for LLMs to master fictional game knowledge that falls outside of their current scope (Cabello et al., 2023). To date, there are over 1,000 Pokémon species and 900 battle moves (bul, 2024b;a), offering a wealth of game knowledge and a large amount of combination possibilities, making the game highly dynamic. Furthermore,

| Environment | Imperfect Info. | Knowledge | Strategic | Adversarial | Dynamic | Game⇒Text |
|---|---|---|---|---|---|---|
| ALFWorld (Shridhar et al., 2020) | ✓ | Low | Low | ✗ | Low | Lossless |
| ScienceWorld (Wang et al., 2022a) | ✓ | Low+ | Low+ | ✗ | Low | Lossless |
| BabyAI (Maxime et al., 2018) | ✗ | Low | Low | ✗ | Low | Lossless |
| OverCooked-AICarroll et al. (2019) | ✗ | Low | Low+ | ✗ | Medium | Lossy |
| Crafter (Hafner, 2021) | ✓ | Medium | Medium | ✗ | High | Lossy |
| Minecraft (Mojang Studios) | ✓ | High | Medium | ✗ | High | Lossy |
| StarCraft II (Vinyals et al., 2017) | ✓ | High | High | ✓ | High | Lossy |
| PokéLLMon (Ours) | ✓ | High | High | ✓ | High | Lossless |

Table 1: Comparison with popular environments in LLM research across several aspects: **Imperfect Info:** The important game information is partially observable. **Knowledge:** The level of knowledge that beyond the scope of LLMs; **Strategic:** The strategic level of playing the game; **Human:** Whether human players can be involved; **Adversarial**: Whether it is an adversarial game or not; **Game⇒Text**: whether translating the game into text is lossy or lossless.

the adversarial feature creates an intense gameplay experience with a high ceiling for reasoning, especially in the presence of powerful opponents such as human experts.

This paper conducts comprehensive evaluation of existing LLMs on game knowledge prediction and Pokémon battles. Empirical evaluations demonstrate that LLMs, especially open-source LLMs, suffer from a severe lack of game knowledge and thus struggle to generate effective actions, indicating our benchmark is a good testbed for grounding techniques. To ground LLMs in games, we first evaluate the impact of game knowledge for different LLMs. Experiments show that the improvement of knowledge is highly related to the inherent reasoning abilities of LLMs, suggesting that when the game knowledge is not the bottleneck, reaching a higher-level performance requires good reasoning ability. Further, we evaluate grounding with self-evolution techniques, which shows that the adversarial setting of PokéLLMon offers an ideal curriculum for learning from self-play experience.

In intense adversarial games, action consistency is an important indicator of performance. Through our analysis, we discover that existing reasoning approaches such as Chain-of-Thoughts (Wei et al., 2022) (CoT) and Reflexion (Shinn et al., 2023) can both lead to inconsistent actions, *i.e.*, the LLM frequently switches Pokémon in consecutive steps and wastes chances to attack. To this end, we introduce an approach that recursively refines the thoughts from the previous steps and can significantly enhance action consistency and gameplay performance. Finally, to demonstrate higher-level reasoning challenges, we test the best-performing LLM in battles against invited human players. The evaluation shows that the LLM player struggles to overcome human players' attrition and misdirection strategies, suggesting a significant room for improvement in reasoning approaches.

In summary, this paper makes three contributions:

- We introduce PokeLLMon, a benchmark that poses new challenges for grounding and reasoning in an environment which features abundant fictional game knowledge and strategic gameplay.

- We conduct a comprehensive evaluation of LLMs' gameplay performance, suggesting that their lack of game knowledge and struggle in Pokémon battles. We further investigate grounding techniques that leverage game knowledge and self-play experience.

- We provide a thorough analysis of reasoning approaches from a new perspective of action consistency, and introduce an effective approach to improve consistency. Additionally, we present high-level reasoning challenges when competing against disciplined human players.

## 2 RELATED WORK

**Interactive environments:** The complexity of environments is depend on two key dimensions: the volume of game knowledge and the strategic depth required. Table 1 summarizes popular environments *w.r.t.* several aspects. Environments that are widely used in recent LLM research were introduced before the rise of LLMs, such as ALFWorld (Shridhar et al., 2020), ScienceWorld (Wang et al., 2022a), and BabyAI (Maxime et al., 2018), featuring limited scenarios or fixed task settings and mainly relying on commonsense knowledge to play. Although the open-world characteristics of Crafter (Hafner, 2021) and Minecraft (Mojang Studios) make them good sandboxes for exploration and crafting tasks, they are not designed as intense adversarial games that require strategic reasoning.

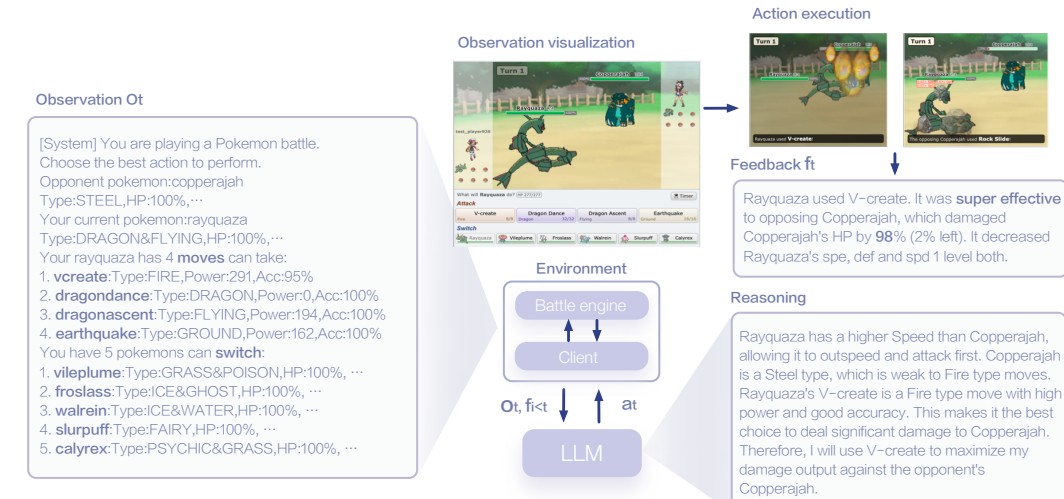

Figure 1: Illustration of how LLMs interact with the environment: At the current time step $t$, the environment outputs an observation prompt $o_t$ that describes the observable information of the current battle state, and the feedback $f_{t-1}$ that describes the action execution outcome of the last time step. The LLM then takes $o_t$ and previous $f_i$ $(i < t)$ as input, conducts reasoning to formulate strategies, and returns an action $a_t$ to the environment. Within the environment, a client is response for translating the text-based $o_t$ and $a_t$ into symbolic information and communicate with the game engine, while the game engine is responsible for generate the next game state based on the current game state and received actions from both players.

StarCraft II (Vinyals et al., 2017), a real-time strategy game, presents challenges for LLMs due to the demand for intensive controls and the difficulty of representing vision-based game states in text (Ma et al., 2023). In comparison, PokéLLMon, which encompasses a high volume of game knowledge and offers strategic gameplay in a format friendly to LLMs (does not require intense control and can be directly translated into text), is an ideal testbed for LLM research.

**LLMs in games:** There are primarily three categories of LLM game agents: (1) Prompt-based approaches that leverage the reasoning abilities of LLMs and feedback from environments, enabling LLMs to iteratively refine strategies. ReAct (Yao et al., 2022) (ALFWorld) introduces the thinking step as a proxy for sub-tasks. Reflexion (Shinn et al., 2023) (ALFWorld) and DEPS (Wang et al., 2023d) (Minecraft) generate self-reflection/explanation based on failure signals and reuse these thoughts for the next trial; In Minecraft, Voyager (Wang et al., 2023a), JARVIS-1 (Wang et al., 2023c) and GTIM (Zhu et al., 2023) iteratively re-generate action code using error messages; (2) Supervised fine-tuning-based methods that collect high-quality trajectories to fine-tune LLMs: E2WM (Xiang et al., 2024) collects embodied experience (VirtualHome (Puig et al., 2018)) using Monte Carlo Tree Search; LLAMARider (Feng et al., 2023) (Minecraft) gathers experience through self-reflection with feedback; (3) Reinforcement learning: GLAM (Carta et al., 2023) (BabyAI-Text) and TWO-SOME (Tan et al., 2024) (OverCooked-AI) discipline LLMs using the PPO algorithm (Schulman et al., 2017).

## 3 ENVIRONMENT

As shown in Figure 1, the environment of PokéLLMon follows a client-server architecture, where the client is implemented as a text interface for LLMs to perceive the game, and the server is an open-source game engine [1] for game execution. The game is synchronized at every step: In each step, both players receive observations and select actions synchronously, and the game engine processes these actions to calculate the next step of the game state. In a battle, each player sends out one Pokémon onto the field, keeping the others off the field for potential switches. The winning condition is to make all the opponent's Pokémon faint by reducing their Hit Points (HP) to zero.

---

[1]https://github.com/smogon/Pokémon-showdown, licensed under the MIT License.

Table 2: Evaluation of LLMs on type effectiveness prediction

| Choice | A. Super effective (51 samples) | | | C. Ineffective (59 samples) | | | D. No effect (8 samples) | | | Overall |
|---|---|---|---|---|---|---|---|---|---|---|
| LLM | Precision | Recall | $F_1$ | Precision | Recall | $F_1$ | Precision | Recall | $F_1$ | $F_1$ |
| Mistral-7B | 0.5000 | 0.0588 | 0.1053 | 0.2190 | 0.3770 | 0.2771 | 0.0000 | 0.0000 | 0.0000 | 0.1841 |
| Gemma-7B | 0.1672 | 1.0000 | 0.2865 | 0.0000 | 0.0000 | 0.0000 | 0.0000 | 0.0000 | 0.0000 | 0.1238 |
| LLaMA3-8B | 0.1818 | 0.1569 | 0.1684 | 0.1871 | 0.5246 | 0.2759 | 0.1333 | 0.2500 | 0.1739 | **0.2225** |
| LLaMA2-7B | 0.1589 | 1.0000 | 0.2742 | 0.0000 | 0.0000 | 0.0000 | 0.0000 | 0.0000 | 0.0000 | 0.1185 |
| LLaMA2-13B | 0.1721 | 0.4118 | 0.2428 | 0.1500 | 0.3443 | 0.2090 | 0.0000 | 0.0000 | 0.0000 | 0.2094 |
| LLaMA2-70B | 0.1902 | 0.7647 | 0.3047 | 0.1613 | 0.1639 | 0.1626 | 0.0000 | 0.0000 | 0.0000 | 0.2130 |
| GPT-3.5-turbo | 0.3778 | 0.3333 | 0.3542 | 0.1944 | 0.8033 | 0.3131 | 0.3333 | 0.2500 | 0.2857 | 0.3290 |
| GPT-4 | 0.9787 | 0.9020 | 0.9388 | 0.7273 | 0.7869 | 0.7559 | 0.7273 | 1.0000 | 0.8421 | 0.8408 |
| GPT-4o | 0.9804 | 0.9804 | 0.9804 | 0.7000 | 0.8033 | 0.7481 | 0.7273 | 1.0000 | 0.8421 | **0.8549** |
| GPT-4o-mini | 0.5652 | 0.2549 | 0.3514 | 0.3750 | 0.5902 | 0.4586 | 0.4000 | 0.7500 | 0.5217 | 0.4172 |

**Observation:** Pokémon battles are an imperfect information game, suggesting that the game state is partially observable. The observation includes the information (stats, abilities, and moves) of a player's own Pokémon team and partial information of the opponent's team, *i.e.*, the species and HP stats of appeared Pokémon. Although the observation is represented as an image from human perspective, it can be symbolically represented without loss of information. At time step $t$, the client translates a symbolic observation into a text observation $o_t$, as shown in Figure 1.

**Action:** There are two types of actions can be chosen: (1) use one of four battle moves, where a move can cause instant damage or produce special effects. The priority of taking moves is determined by the current speed of two Pokémon on the field. (2) Switch to an off-the-field Pokémon. If choose to switch, the switching will happen immediately before the opposing Pokémon take a move, and the switch-in Pokémon cannot take an extra action in this step. The admissible actions (up to 9 choices, *i.e.*, 4 moves plus 5 switch-in options) are included in $o_t$. The LLM is adopted as the policy model $\pi_\theta$ to generate reasoning and an action $a_t$.

**Feedback:** The action will then be executed and during execution, human players perceive the feedback from the battle animation that reflects the effectiveness of chosen actions. To compensate this information, we introduce four types of textual feedback: (1) The change in HP. (2) The outcome of actions in terms of type-effectiveness, *i.e.*, whether it is super-effective, ineffective, or has no effect. (3) The priority of move execution. (4) The effects of special moves on stats/status changes, weather, and side conditions. The feedback $f_i$ ($i < t$) is included in the $o_t$ for LLMs to perceive its previous actions and outcomes. In the environment, we also provide numeric reward value derived based on the outcome of actions.

## 4 PRELIMINARY EVALUATION

### 4.1 GAME KNOWLEDGE EXAMINATION

Type-effectiveness is fundamental knowledge in Pokémon battles. It defines the effectiveness of a certain type of Pokémon when attacked by a certain type of attack. For example, a Fire-type Pokémon is vulnerable to Water-type attacks (see the chart in Appendix D). We use the chart to generate multiple-choice questions to test the game knowledge of LLMs. The question template is as follows:

**Multi-choice question**: In Pokémon battles, a $type_1$ attack is ___ against a $type_2$ Pokémon.
A. Super-effective (2x)    B. Standard (1x)    C. Ineffective (0.5x)    D. No effect (0x)

Table 2 shows the results of existing LLMs, where we report the precision, recall, and F1 score for choices A, C, and D (with B being the majority answer), as well as the weighted F1 score across the three choices. We observe that the overall F1 score is low, especially for open-source LLMs, suggesting that game knowledge is largely beyond the scope of pre-trained LLMs. Even though the best-performing LLM, GPT-4o, achieves quite accurate results, we will show that this is still insufficient in intense battles where a single mistake can lead to a significant disadvantage.

### 4.2 GAME PERFORMANCE EVALUATION

Table 3 shows the gameplay performance of existing LLMs against ExperSystem, an expert system that simulates human's decision-making process with numeric damage calculation (details in Ap-

Table 3: Evaluation of the gameplay performance of LLMs ($n$ is the number of few-shot examples)

| Shot # | | n=0 | | | n=1 | | | n=3 | |
|---|---|---|---|---|---|---|---|---|---|
| **Player** | **Win Rate↑** | **Battle Score↑** | **Error Rate↓** | **Win Rate↑** | **Battle Score↑** | **Error Rate↓** | **Win Rate↑** | **Battle Score↑** | **Error Rate↓** |
| ExpertSystem | 0.5000 | 6.000 | 0.000 | - | - | - | - | - | - |
| Human | **0.5984** | **6.750** | 0.000 | - | - | - | - | - | - |
| Random | 0.0120 | 2.340 | 0.000 | - | - | - | - | - | - |
| MaxPower | 0.1040 | 3.790 | 0.000 | - | - | - | - | - | - |
| Mistral-7B | 0.0486 | 3.047 | 0.008 | - | - | >0.90 | - | - | >0.90 |
| Gemma-7B | 0.0866 | 3.684 | 0.022 | - | - | >0.90 | - | - | >0.90 |
| LLaMA3-8B | 0.1034 | **3.902** | 0.028 | - | - | >0.90 | - | - | >0.90 |
| LLaMA2-7B | 0.0760 | 3.572 | 0.015 | - | - | >0.90 | - | - | >0.90 |
| LLaMA2-13B | 0.0829 | 3.610 | 0.006 | - | - | >0.90 | - | - | >0.90 |
| LLaMA2-70B | **0.1065** | 3.703 | 0.031 | - | - | >0.90 | - | - | >0.90 |
| GPT-3.5-turbo | 0.1351 | 4.089 | 0.000 | 0.0988 | 3.673 | 0.000 | 0.1333 | 4.027 | 0.000 |
| GPT-4o-mini | 0.0719 | 3.127 | 0.000 | 0.1236 | 3.789 | 0.000 | 0.1634 | 4.254 | 0.000 |
| GPT-4o | **0.3100** | 4.719 | 0.000 | 0.2917 | 4.605 | 0.000 | 0.2986 | 4.578 | 0.001 |
| GPT-4 | 0.2673 | **4.954** | 0.000 | 0.2644 | 4.926 | 0.000 | 0.2692 | 4.603 | 0.000 |

pendix A). In each step, LLMs take $o_t$ as input and output $a_t$ without explicit reasoning (see Section 6 for the evaluation of reasoning). $n$ is the number of few-shot examples used in prompt. To provide more comparisons, we report the performance of Human (randomly-matched human players from game server[2]), Random (method that randomly selects actions) and Max-Power (method that selects moves with the highest move power). Each experiment is run over 1,000 times for open-source LLMs, and over 200 times for GPTs. The error rate represents the proportion of inadmissible actions generated by LLMs. The battle score is defined as follows:

$$\text{Battle Score} = \sum_{p \in \mathcal{P}} \text{HP}(p) + \sum_{p \in \mathcal{O}} (1 - \text{HP}(p)) \tag{1}$$

where $\text{HP}(p)$ is the percentage of Pokémon $p$'s HP at the end of a battle, $\mathcal{P}$ and $\mathcal{O}$ are the player's team and the opponent's team. The score is the sum of the remaining HP percentages of the player's Pokémon and the HP loss percentages of the opponent's Pokémon, rewarding both preservation of health and infliction of damage. For LLMs, the temperature $\tau$ is set to 0 to reduce inconsistency.

From Table 3, we make three observations: (1) all the LLMs, especially open-source LLMs, perform badly in Pokémon battles, and their gameplay performance is positively correlated with their performance in type-effectiveness prediction. (2) With few-shot examples, open-source LLMs repeat the actions in examples rather than generation, making the output inadmissible (error rate >90%). We conjecture that is because they are less aligned with human instructions compared to GPTs, making them difficult to generalize to cases they have never seen. (3) Few-shot examples do not bring any improvement for GPT-4o and GPT-4. This is because that Pokémon battles are dynamic and involve numerous possibilities, making it difficult for few-shot examples to cover all situations.

## 5 GROUNDING

In this section, we introduce two ways of grounding LLMs in games: grounding with knowledge and with self-play experience.

### 5.1 GROUNDING WITH KNOWLEDGE

In PokéLLMon, we provide interfaces for knowledge retrieval during battles. The game knowledge is crawled from Pokémon Wiki (pok) and Bulbapedia (bul), and structured in key-value pairs. Specifically, we adopt two essential categories of game knowledge: (i) Type-effectiveness, which describes the strengths and weaknesses of both the attack type and the Pokémon type, for example:

> **Key:** Fire-type Pokémon
> **Value:** Is resistant to Fire, Grass, Ice attacks, and is vulnerable to Water, Ground, Rock attacks.

(ii) Move effect, which describes the effects of moves, for example:

---

[2]Playing with humans follows the bot usage policy of the game content provider.

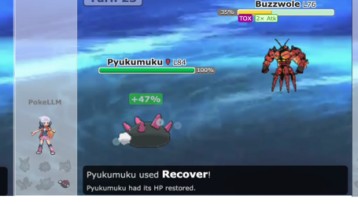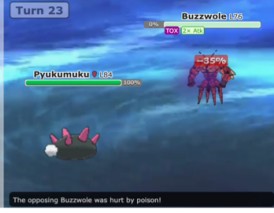

| (1) LLM uses the Toxic move to poison the opposing Pokemon | (2) The LLM frequently uses the Recover move to prevent its Pokémon from fainting | (3) The opposing Pokémon is depleted by the poisoning damage |

Figure 2: With game knowledge, the LLM exhibits an attrition strategy using two moves: It first uses *Toxic* to poison the opponent, which inflicts additional poisoning damage on every turn. Then, it prolongs the battle by frequently healing itself with *Recover*, a move that can restores 50% of HP. As a result, the opponent gradually weakens due to the poisoning damage and faints after 7 turns.

> **Key:** Toxic
> **Value:** Toxic poisons the target, causing it to lose progressively increasing HP each turn.

During games, the agent retrieves the type-effectiveness knowledge using the opponent Pokémon's type, and move knowledge using the move name of its own Pokémon. Retrieved knowledge will be integrated into the observation $o_t$ as the input for LLM to generate action $a_t$.

Table 4: Evaluation on the impact of game knowledge

| LLM | Win Rate | W.R. w/ know. | Battle Score | B.S. w/ know. | $F_1$ |
|---|---|---|---|---|---|
| GPT-3.5-turbo | 0.1351 | 0.0795 (-5.56%) | 4.089 | 3.927 (-0.162) | 0.3290 |
| GPT-4o-mini | 0.0719 | 0.1584 (+8.65%) | 3.127 | 3.695 (+0.568) | 0.4172 |
| GPT-4o | **0.3100** | 0.4217 (+11.17%) | 4.719 | 5.413 (+0.694) | **0.8549** |
| GPT-4 | 0.2673 | **0.4744** (**+20.71%**) | **4.954** | **5.869** (**+0.915**) | 0.8408 |

In Table 4 we evaluate the impact of game knowledge in battles against ExpertSystem, where $F_1$ is the overall F1 score of type-effectiveness prediction task. Among four LLMs, game knowledge significantly enhances GPT-4, with an increase of 20.71% in win rate, while it even hurts the performance of GPT-3.5-turbo, leading to a win rate drop of 5.56%. This suggests that improvement in knowledge is related to the inherent reasoning ability of LLMs. Moreover, by comparing GPT-4o with GPT-4, we can conclude that when game knowledge is not the bottleneck, achieving higher-level performance requires good (explicit/implicit) reasoning ability.

With knowledge, we observe that GPT-4 exhibits **emergent behaviors**: As shown in Figure 2, when the agent has a move that can inflict additional damage regularly, such as *Toxic*, and another move that can recover its HP, such as *Recover*, the LLM develops the attrition strategy: It first poisons the opposing Pokémon to cause regular damage every turn and then frequently recovers HP to prevent its Pokémon from fainting. After 7 turns, the opponent's HP is gradually depleted by the poisoning damage until it faints.

## 5.2 GROUNDING WITH SELF-PLAY EXPERIENCE

Adversarial games offer an ideal learning curriculum by allowing self-play against an opponent of the same gameplay level, as demonstrated in AlphaGoZero (Silver et al., 2017) and OpenAI Five (Berner et al., 2019b). We introduce grounding LLMs through learning from self-play experience. Self-evolution can be considered as iterations of two phases: trajectory sampling and fine-tuning. For trajectory sampling, we enable two agents initialized from the same LLM to act as a pair of opponents in Pokémon battles to collect rollouts. In the fine-tuning stage, we adopt two approaches that can be easily integrated into self-play:

- Rejection sampling Fine-Tuning (RFT) (Yuan et al., 2023): RFT samples the trajectories of winners to fine-tune LLMs by maximizing the log-likelihood of actions sampled from winning trajectories. The loss function can be formalized as:

$$L = -\frac{1}{N} \sum_{i=1}^{N} \sum_{t=1}^{T} \log \pi_\theta(a_{i,t}^w | o_{i,t}^w) \quad (2)$$

Table 5: Evaluation of self-evolution techniques. ◑ denotes playing against ExpertSystem and ● denotes playing against MaxPower.

| Tag | Method | Win Rate ↑ | Battle Score ↑ |
|---|---|---|---|
| ◑ | Origin | 0.1075 | 3.904 |
| | RFT | 0.1161 | 4.194 |
| | DPO | 0.1212 | 4.207 |
| ● | Origin | 0.5641 | 6.003 |
| | RFT | 0.5920 | 6.266 |
| | DPO | 0.5984 | 6.302 |

where $(o_{i,t}^w, a_{i,t}^w) \in \tau_i^w$, and $\tau_i^w$ is the trajectory of the winner in $i$-th battle.

- Direct Preference Optimization (DPO) (Rafailov et al., 2024): For the $i$-th battle, we sample pairs of $(o_{i,t}^w, a_{i,t}^w)$ and $(o_{i,t}^l, a_{i,t}^l)$ from winner trajectory $\tau_i^w$ and loser trajectory $\tau_i^l$, then fine-tune $\pi_\theta$ by minimizing the following loss function:

$$L = -\frac{1}{N} \sum_{i=1}^{N} \sum_{t=1}^{T} \left[ \log \sigma \left( \beta \log \frac{\pi_\theta(a_{i,t}^w | o_{i,t}^w)}{\pi_{\text{ref}}(a_{i,t}^w | o_{i,t}^w)} - \beta \log \frac{\pi_\theta(a_{i,t}^l | o_{i,t}^l)}{\pi_{\text{ref}}(a_{i,t}^l | o_{i,t}^l)} \right) \right] \tag{3}$$

where $\pi_{\text{ref}}$ is the reference model initialized using the $\pi_\theta$ from the last iteration to prevent excessive drift. Optimizing the DPO loss is essential to increase the likelihood margin between the winner actions and loser actions.

The implementation details can be found at Appendix A. From Table 5 we observe that both RFT and DPO obtains better performance than the original LLM player, suggesting that LLMs are able to learn in environments with self-play experience. The second observation is that DPO outperforms RFT. This is because RFT only learns from the trajectories of winners, while DPO minimizes the likelihood of losers' actions, making the learning more effective. Overall, adversarial self-play makes self-evolution efficient: If it is not a self-play setting, RFT will be difficult to obtain winner trajectories when playing against a powerful opponent, or quickly get saturated when playing against a very weak opponent.

## 6 REASONING

### 6.1 EVALUATION OF REASONING METHODS

In intense adversarial games, inconsistent actions often lead to defeat by wasting opportunities to make strategic moves. To benchmark existing reasoning approaches, we introduce two additional metrics that measure the degree of action inconsistency and are closely linked to gameplay performance.

**Metrics:** In Pokémon Battles, there are two categories of steps: (1) force switch step that is compulsory to switch when the pokemon on-the-field faints and the player decides which pokemon off-the-field to switch in; (2) active step, *i.e.*, to take move or switch Pokemon. Switching in an active step provides advantages, such as having a Pokémon with type effectiveness over the opposing Pokémon, however, it also means to give the opposing Pokemon a free turn to take a move (and abandon the boost of stats if have). Frequent switching can waste opportunities to attack and will lead to defeat. Therefore, we use switch rate and consecutive switch rate to measure the action consistency of a player, formalized as follows:

$$\text{Switch Rate} = \frac{\text{\# of active switch}}{\text{\# of active step}} \tag{4}$$

$$\text{Consecutive Switch Rate} = \frac{\text{\# of consecutive active switch}}{\text{\# of active switch}} \tag{5}$$

Switch rate measures the proportion of times a player makes an active switch, and consecutive switch rate measures the frequency with which a player switches in consecutive steps against the same opponent.

**Approaches**: As shown in Figure 3, we evaluate reasoning approaches, including IOPrompt, Chain-of-Thoughts (Wei et al., 2022), Self Consistency (Wang et al., 2022b) (SC-CoT), Tree-of-Thoughts (Yao et al., 2023), Relexion (Shinn et al., 2023) and Last-Thoughts (See Appendix A for more descriptions).

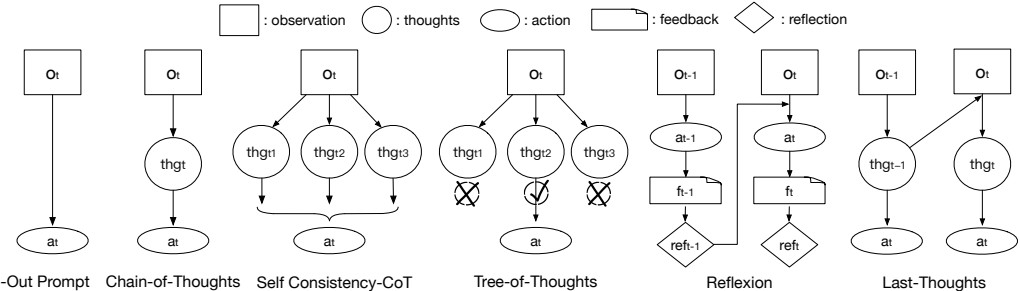

Figure 3: Illustration of reasoning approaches evaluated in Table 6.

Table 6: Evaluation of reasoning approaches *w.r.t.* gameplay performance and action consistency.

| Method | Win Rate↑ | Battle Score↑ | Switch Rate | Con. Switch Rate↓ |
|---|---|---|---|---|
| IOPrompt | 0.4217 | 5.413 | 0.3356 | 0.2442 |
| CoT | 0.3713 | 5.127 | 0.3344 | 0.2647 |
| SC-CoT | 0.4065 | 5.037 | 0.3643 | 0.0954 |
| ToT | 0.2549 | 4.398 | 0.3163 | 0.2938 |
| Reflexion | 0.2923 | 5.015 | 0.3680 | 0.2982 |
| LastThoughts | **0.4667** | **5.840** | 0.2227 | **0.0861** |
| IOPrompt ($\tau$=0.0) | 0.4217 | 5.413 | 0.3356 | 0.2442 |
| IOPrompt ($\tau$=0.5) | 0.3818 | 5.285 | 0.3204 | 0.2504 |
| IOPrompt ($\tau$=1.0) | 0.3019 | 4.937 | 0.3224 | 0.2689 |

We adopt GPT-4o [3] as the LLM and game knowledge, and run every experiments over 200 trials. Two one-shot examples (one for active step and one for force switch step) are designed to guide reasoning. Temperature $\tau$ is set to 0 to reduce inconsistency besides for SC-CoT, which requires to encourage different reasoning, we set $\tau$ to 0.5. For SC-CoT and ToT, the number of branch $b$ is set to 3.

**Analysis:** Table 6 reports the gameplay performance with reasoning approaches. Surprisingly, we observe that approaches like CoT, ToT and Reflexion significantly decreases the win rate and battle score compared to the vanilla IOPrompt without any reasoning. According to the metrics, the drop of game performance can be attributed to the increase in consecutive switch rate, suggesting that with reasoning, the agent is tend to switch in a new pokemon and switch it out in the next turn, without taking moves. Below, we break down the analysis for every approach.

**CoT**: Why does CoT lead to action inconsistency? Let us consider two cases: In the first case, our Pokémon is facing an opponent, and in the second case, everything is the same except that the opponent has boosted its attack stats twofold. For IOPrompt, the difference between two cases is only two extra tokens indicating the boosted stats. However, with reasoning, the LLM will describe the disadvantage of facing a boosted opponent in its thoughts. From a generation perspective, the thoughts increase the discrepancy between two observations in the representation space, making the LLM more likely to generate different actions. From a gameplay perspective, conditioned on these thoughts, the LLM tends to protect its Pokémon from fainting by switching it out of the battlefield, yet neglects the fact that consecutive switching actually wastes the chance to attack.

**SC-CoT and ToT**: As show in Table 6, Compared to CoT, SC-CoT decreases the consecutive switch rate while increasing the switch rate, which suggests that SC-CoT is more consistent in consecutive steps with similar observations, yet more likely to switch Pokémon when facing different opposing Pokémon. This is because the majority voting reduces the inconsistency brought by the randomness of sampling. However, it enhances the probability of switching when switching is the predominant option in the probabilistic distribution conditioned on thoughts. A competitor, ToT, replacing the majority voting of SC-CoT by self-evaluation, although outperforms SC-CoT in easily-evaluated tasks like Game of 24 (Yao et al., 2023), works even worse than CoT and SC-CoT in our benchmark. This is because the LLM is unable to make correct self-evaluation due to its lack of game experience, and thus tends to agree with the proposals, which does not provide any benefit for reducing inconsistency, and sometimes even prioritizes the sub-optimal actions, leading to a drop of performance.

**Last-Thoughts**: We introduce a simple yet effective solution: Last-Thoughts. By recursively taking the thoughts from the last step into current reasoning procedure, we observe a significant drop of

---

[3]GPT-4o is more cost-efficient than GPT-4, allowing us to run more experimental trials for accurate estimation.

consecutive switch rate from 34.50% to 9.31%, which thereby boosts the win rate from 35.00% to 46.67%. With Last-Thoughts, the LLM refines its thoughts based on its previous thoughts instead of generating from scratch. For the perspective of generation, last thoughts can be deemed as a summary of the last observation, thus two consecutive observations become more similar in the representation space, leading to consistent outputs.

**Reflexion**: As shown in Figure 3, the LLM uses reflections on the previous step to generate a new action. There are two reasons that reflection does not work well: (1) unlike static task settings (Shridhar et al., 2020), Pokémon battles are dynamic, suggesting that reflections from the past are likely outdated; (2) Even if not outdated, the LLM tends to think that the last action did not meet its expectations, and thus chooses to switch to a new Pokémon to attack, leading to the increase of inconsistency. However, we believe reflection can be beneficial in a dynamic environment if adjusted appropriately, *i.e.*, by generating reflections offline and retrieving them based on similarity during battles. As a result, the problem of outdated reflections can be addressed, and the agent also avoids wasting chances in a trial-and-error style.

**Impact of temperature on consistency**: Temperature $\tau$ controls the sharpness of distribution for sampling tokens. In Table 6 we report the performance of IOPrompt *w.r.t.* temperatures varying from 0 to 2, suggesting that a lower temperature can reduce inconsistency and increase performance.

> **Findings:** In an intense adversarial game, action consistency is an important indicator for gameplay performance. The inconsistency introduced by CoT is due to the increased discrepancy between consecutive observations, and can be reduced by integrating the previous thoughts in the next step reasoning (Last-Thoughts). Furthermore, a lack of game experience and the highly dynamic feature undermine approaches relying on self-evaluation such as ToT and Reflexion.

## 6.2 Reasoning Challenges in Online Battles

Due to the characteristics of adversarial games, the difficulty of reasoning can be further increased by playing against powerful opponents, *e.g.*, human experts. In order to demonstrate higher-level reasoning challenges, we set up online battles with human players randomly matched from the game server. Online games adhere to the bot usage principles of the game content provider and follow the principle of disclosure: human players are informed through the invitations that they are playing against a non-human agent, and battles only begin upon acceptance.

Table 7: Performance on online battles against human players.

| v.s. Player | Win Rate ↑ | Battle Score ↑ |
|---|---|---|
| Ladder Player | 0.4857 | 5.76 |

Table 7 shows the results of an enhanced agent [4] over 100 online battles against human players matched from the ladder competition. We then introduce two reasoning challenges observed when playing against human players, namely long-term dilemma and Theory-of-Mind inference.

### 6.2.1 Long-term dilemma

We observe that the current LLM agent still lacks of long-term planning ability, *i.e.*, it tends to achieve short-term goals like instant damage, and thus is vulnerable to the dilemma that requires long-term strategy to break. A good case to measure the long-term planning ability is the human players' attrition strategy, which frequently recovers the Pokémon's HP and gradually deplete the opponent's HP with occasional attacks or regular damage moves such as Toxic.

Table 8 reports the performance of the agent in battles where human players either use the attrition strategy or not. We can observe that the agent lost the majority of games when humans performed the attrition strategy, where,the turn number of battles is significantly increased. The key to breaking the dilemma is to form a plan that cross multiple steps: firstly boost its Pokémon's attack to a high stage and then attack to cause unrecoverable damage, which is a long-term goal that requires joint efforts across many steps.

---

[4]To make the agent capable of playing against human, we include more knowledge including ability/item/-condition, and adopt GPT-4 as the policy, making it reaches a win rate of 60% against ExpertSystem.

Table 8: Battle performance impacted by the attrition strategy

| Battles | Win rate ↑ | Score ↑ | Turn # | Battle # |
|---------|-----------|---------|--------|----------|
| w. Attrition | 0.1875 | 4.29 | 33.88 | 16 |
| w/o Attrition | 0.5393 | 6.02 | 15.95 | 89 |

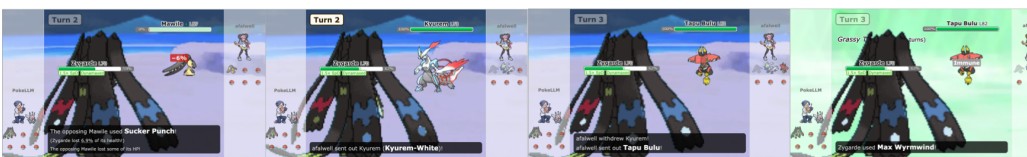

(1) Opposing pokemon fainted, trigerring a force switch  (2) Opponent switch in a pokemon vulnerable to Dragon type attack  (3) The LLM chooses to use Dragon type attack, the opponent choose to switch  (4) Our Pokémon uses Dragon type attack, no effect to the opposing pokémon

Figure 4: An experienced human player misdirects the LLM to waste an attack chance: In a force switch, the opponent switched in a Pokémon that has a type weakness to agent's current Pokémon's Dragon-type attack. Naturally, the agent chose to use a Dragon-type attack, while the opponent chose to switch in another Pokémon that is immune to Dragon-type attacks. Since the switching occurs before the attack, the Dragon-type attack chosen by the agent has no effect on the opponent's switch-in Pokémon, resulting in a waste of an enhanced attack chance.

### 6.2.2 THEORY-OF-MIND INFERENCE

Theory-of-Mind (ToM) (Frith & Frith, 2005) thinking involves inferring others' intentions from a shifted perspective, which demonstrate enhancement in imperfect information games (Guo et al., 2023) and cooperation games (Zhang et al., 2023; Agashe et al., 2023). In battles, we observe experienced human players misdirected the LLM to bad actions, because it lacks of ToM thinking and makes decisions solely based on the observation.

As shown in Figure 4, the Pokémon from our side has one chance to use an enhanced attack move. At the end of turn 2, the opposing *Mawile* faints, leading to a force switch, and the opponent player chooses to switch in another Pokémon for the next turn. This is a trick to lure the agent to use a dragon-type move, given that dragon-type attack is super effective to the opposing Pokémon. As expected, the agent chooses to use the dragon-type move. However, before our Pokémon attacks, the opponent switches in another Pokémon that is immune to dragon-type attacks, causing our enhanced attack opportunity wasted.

To summarize, attrition and misdirection strategies by human players pose higher-level reasoning challenges for LLMs. Due to the adversarial gameplay setting and the presence of powerful opponents, the benchmark presents a high ceiling for strategic reasoning.

## 7 CONCLUSION

This paper introduces a new grounding and reasoning benchmark for LLMs in Pokémon Battles, featuring rich game knowledge and a high ceiling for strategic reasoning due to its highly dynamic, intense and adversarial gameplay, especially in the presence of professional opponents. Through experiments we verify the lackness of game knowledge and experience of existing LLMs, investigate the potential grounding solutions with game knowledge and self-play experience. With a thorough analysis, we identify the weakness of existing reasoning approaches from a new perspective of action consistency, and provide an effective solution to reduce inconsistency. Finally, we introduce two advanced strategies exhibited by human players that pose higher-level reasoning challenges. Overall, we believe PokéLLMon is an ideal testbed for developing grounding and reasoning techniques.

## 8 ETHICS STATEMENT

The goal of the paper is only for AI development akin to previous benchmarks in StarCraft II (Vinyals et al., 2017) and DOTA (Berner et al., 2019a). We acknowledge that our developments follows the bot usage principle of the service provider[5]. To further address ethical concerns, the environment is implemented in compliance with the principle of disclosure, *i.e.*, for online games, human players are informed that they are playing with non-human players, and games only begin upon acceptance.

---

[5] https://gist.github.com/Kaiepi/becc5d0ecd576f5e7733b57b4e3fa97e

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

## A    METHOD DESCRIPTIONS

**ExpertSystem**: ExpertSystem simulates a human player's decision-making procedure. The approach includes a state evaluation function that calculates the battle advantage between the player's Pokémon and the opponent's, taking into account factors such as type-effectiveness, current HP and stats of Pokémon of both sides. If the current situation is favorable, the approach calculates the damage for attacks by considering move power, type effectiveness, and the stats of the Pokémon, and finally selects the move with the highest damage value; If the current situation is unfavorable and no strategic moves are available, it evaluates whether switching to another Pokémon could lead to a more advantageous situation using state evaluation function.

Overall, ExpertSystem is programmed with a decision-making procedure with numeric damage calculation, enabling it to generate effective actions for both move selection and switching, serving as a competitive opponent in our experiments.

**RFT and DPO fine-tuning**: We first supervisedly fine-tune LLaMA2-7B on 10,240 sampled frames (steps) of the expert system to ensure it 100% outputs admissible actions, as the initial LLM. We run the self-evolution iteration for 5 times, and in each iteration, we sample 100,000 frames (steps). The batch size for training is set to 64, and each sample is trained once. The learning rate is set to 1e-5 for RFT and 1e-6 for DPO based on empirical hyper-parameter tuning. RMSProp is adopted as the optimizer. For DPO, $\beta$ is set to 0.1.

**Reasoning approaches**:

- In-Out Prompt (IOPrompt): At each time step, LLM takes the observation as the input and directly output an action. We do not use few-shot examples as they do not provide obvious improvement (Table 3).

- Chain-of-Thoughts (CoT) (Wei et al., 2022): At each step, LLM generates thoughts to analyze the battle-field, then outputs an action conditioned on the thoughts. The reasoning includes the comparison of stats, type-effectiveness and evaluation of moves.

- Self Consistency (SC-CoT) (Wang et al., 2022b): At each step, LLM independently generates $b$ reasoning branches and do majority voting for the final output action.

- Tree-of-Thoughts (ToT) (Yao et al., 2023): At each step, LLM analyzes the battlefield and proposes $b$ top actions. LLM then criticizes and evaluates these actions with reasoning and selects the best action to output.

- Reflexion (Shinn et al., 2023): LLM reflects on the outcome of the action $a_{t-1}$ taken in the previous step, and use the reflection to generate action $a_t$ in the next time step.

- Last-Thoughts: At each step, LLM generates CoT reasoning by taking into consideration the thoughts from the last step.

## B    PROMPT DESCRIPTIONS

### B.1    IOPROMPT

**Input**: observation $o_t$ (game knowledge is presented in bold)

> [System] You are playing a Pokemon battle and the goal is to win. Select the best action and output.
> Historical turns:
> Turn 12: Bouffalant used Megahorn, which was super effective to opposing Reuniclus. It damaged opposing Reuniclus's HP by 59% (41% left). Opposing Reuniclus used Focus Blast, which was super effective to Bouffalant. It damaged Bouffalant's HP by 53% (0% left). Bouffalant fainted. Bouffalant outspeeded opposing Reuniclus. You sent out Doublade.
> Current turn:
> Opponent has 2 pokemons left.
> Opponent current pokemon:reuniclus,
> Type:PSYCHIC,HP:41%,Atk:157,Def:174,Spa:258,Spd:191,Spe:99

> **reuniclus as defender, BUG,GHOST deal 2x damage; PSYCHIC,FIGHTING only deal 0.5x damage to reuniclus**
> **reuniclus as attacker, PSYCHIC deal 2x damage to FIGHTING pokemon; PSYCHIC deal 0.5x damage to PSYCHIC,STEEL pokemon**
> Your current pokemon:doublade,
> Type:STEELGHOST,HP:10%,Atk:228,Def:293,Spa:121,Spd:128,Spe:105
> Your doublade has 4 moves can take:
> swordsdance:Type:NORMAL,Cate:Status,Power:0,Acc:100%,
> **Effect:Raises the user's Attack by two stages.**
> ironhead:Type:STEEL,Cate:Physical,Power:105,Acc:100%,
> **Effect:Has a 30% chance to make the target flinch.**
> closecombat:Type:FIGHTING,Cate:Physical,Power:157,Acc:100%,
> **Effect:Lowers the user's Defense and Special Defense by one stage after inflicting damage.**
> shadowsneak:Type:GHOST,Cate:Physical,Power:52,Acc:100%,
> **Effect:Inflicts regular damage with no additional effect.**
> You have 2 pokemons can switch:
> azelf:Type:PSYCHIC,HP:0%,Atk:249,Def:160,Spa:249,Spd:160,Spe:233,
> Moves:[uturn,BUG],[psychic,PSYCHIC],[fireblast,FIRE]
> tsareena:Type:GRASS,HP:100%,Atk:250,Def:213,Spa:132,Spd:213,Spe:169,
> Moves:[rapidspin,NORMAL],[tripleaxel,ICE],[powerwhip,GRASS]

**Output**: action $a_t$

> **Action $a_t$**: Move: shadowsneak

## B.2 CHAIN-OF-THOUGHTS (COT) & SELF-CONSISTENCY (SC-COT)

CoT and SC-CoT both use the same prompt template, except that SC-CoT outputs multiple times independently and performs majority voting.

**Input**: observation $o_t$ (game knowledge is presented in bold)

> [System] You are playing a Pokemon battle and the goal is to win. Below is an example to teach you how to reason and select an action.
> ===Example Start===
> ...
> ===Example End===
> Below is the real case:
> Historical turns:
> Turn 20: Opposing Salazzle used Flamethrower, which damaged Dusknoir's HP by 12% (0% left). Dusknoir fainted. You sent out Lickilicky.
> Current turn:
> Opponent has 1 pokemons left. Opponent current pokemon:salazzle,
> Type:POISON&FIRE,HP:35%,Atk:152,Def:146,Spa:229,Spd:146,Spe:239
> **salazele as a defender, FIGHTING attack only deal 0.5x damage to it**
> Your current pokemon:lickilicky,
> Type:NORMAL,HP:44%,Atk:195,Def:213,Spa:187,Spd:213,Spe:135
> Your lickilicky has 4 moves can take:
> wish:Type:NORMAL,Cate:Status,Power:0,Acc:100%
> **Effect:User will recover half its max HP at the end of the next turn.**
> bodyslam:Type:NORMAL,Cate:Physical,Power:114,Acc:100%
> **Effect:Has a 30% chance to paralyze the target.**
> knockoff:Type:DARK,Cate:Physical,Power:87,Acc:100%
> **Effect:Target drops its held item.**
> swordsdance:Type:NORMAL,Cate:Status,Power:0,Acc:100%
> **Effect:Raises the user's Attack by two stages.**

You have 2 pokemons can switch:
togekiss:Type:FAIRY&FLYING,HP:28%,Atk:85,Def:198,Spa:238,Spd:230,Spe:174,
Moves:[airslash,FLYING],[aurasphere,FIGHTING]
wobbuffet:Type:PSYCHIC,HP:48%,Atk:69,Def:167,Spa:119,Spd:167.

**Output**:

**Thoughts**: The opponent's Salazzle is faster than my Lickilicky, and it has a high Special Attack stat. Lickilicky's current HP is 44%, which means it might not survive a strong special attack from Salazzle. Switching to Togekiss is risky due to its low HP (28%), and Wobbuffet might not be able to counter effectively due to its lower defenses. Lickilicky's best move is to use Knock Off, which can deal damage and potentially remove Salazzle's held item, reducing its effectiveness. This is a safer option than switching, as it allows Lickilicky to potentially deal damage before fainting.
**Action** $a_t$: Move: knockoff

### B.3   TREE-OF-THOUGHTS (ToT)

For each step, ToT has two phases: proposal and self-evaluation.

**Phase 1: Proposal**

**Input:** observation $o_t$ (game knowledge is presented in bold)

[System] You are playing a Pokemon battle and the goal is to win. Below is an example to teach you how to reason and propose 3 best actions.
===Example Start===
...
===Example Ends===
Here is the real case:
Historical turns: Turn 3: opposing Thievul used Dark Pulse. It damaged Dragalge's HP by 78% (22% left). Dragalge used Focus Blast. It missed. opposing Thievul outspeeded Dragalge.
Current turn:
Opponent has 6 pokemons left.
Opponent current pokemon:thievul,
Type:DARK,HP:52%,Atk:152,Def:152,Spa:609(4 stage),Spd:212,Spe:209
**thievul as defender, FIGHTING deal 2x damage; GHOST,DARK only deal 0.5x damage; PSYCHIC have no effect to thievul**
**thievul as attacker, DARK deal 2x damage to PSYCHIC,GHOST pokemon; DARK deal 0.5x damage to DARK,FIGHTING pokemon**
Your current pokemon:dragalge,
Type:POISON&DRAGON,HP:28%,Atk:135,Def:206,Spa:219,Spd:264,Spe:126
Your dragalge has 4 moves can take:
toxicspikes:Type:POISON,Cate:Status,Power:0,Acc:100%,
**Effect:Scatters poisoned spikes, poisoning opposing Pokémon that switch in.**
dracometeor:Type:DRAGON,Cate:Special,Power:134,Acc:90%,
**Effect:Lowers the user's Special Attack by two stages after inflicting damage.**
sludgewave:Type:POISON,Cate:Special,Power:98,Acc:100%,
**Effect:Has a 10% chance to poison the target.**
focusblast:Type:FIGHTING,Cate:Special,Power:124,Acc:70%,
**Effect:Has a 10% chance to lower the target's Special Defense by one stage.**
You have 5 pokemons can switch:
blacephalon:Type:FIREGHOST,HP:100%,Atk:205,Def:129,Spa:284,Spd:170,Spe:215,
Moves:[fireblast,FIRE],[shadowball,GHOST],[psyshock,PSYCHIC]
jellicent:Type:WATERGHOST,HP:100%,Atk:107,Def:168,Spa:193,Spd:227,Spe:151,
Moves:[shadowball,GHOST],[scald,WATER]
snorlax:Type:NORMAL,HP:100%,Atk:228,Def:154,Spa:154,Spd:228,Spe:96,

Moves:[doubleedge,NORMAL],[darkestlariat,DARK],[facade,NORMAL]
slowbro:Type:WATERPSYCHIC,HP:100%,Atk:132,Def:236,Spa:219,Spd:185,Spe:100,
Moves:[scald,WATER]
regieleki:Type:ELECTRIC,HP:100%,Atk:199,Def:122,Spa:199,Spd:122,Spe:353,
Moves:[explosion,NORMAL],[thunderbolt,ELECTRIC],[voltswitch,ELECTRIC]

**Output**:

**Thoughts**: Thievul's previous Dark Pulse did 78% damage, leaving Dragalge at 22% HP. Since Thievul outspeeds Dragalge, it will KO Dragalge before it can attack. Switching out Dragalge keeps it as fodder or for a potential safe switch-in later. If choose to switch, the switch in pokemon will first bear an attack. With a high SpD, Snorlax can potentially survive a boosted Dark Pulse from Thievul. If Thievul misses, we can use focusblast to knock out Thievul.
**Proposal 1**: Switch: snorlax
**Proposal 2**: Move: focusblast
**Proposal 3**: switch: dracometeor

**Phase 2: Self-evaluation**

**Input:** The input is the observation $o_t$ and the proposal in phase 1, along with an example to teach LLM for self-evaluation. We omit the prompt here due to similarities in content.

**Ouput:**

**Thoughts**: Since Thievul outspeeds Dragalge and has high Spa, it will KO Dragalge before it can attack. If choose to use attack, it is very likely be knocked out in the next step. However, if choose to switch in a non force switch, the switch in pokemon will first bear an attack and be knocked out since Thievul's Special Attack has been boosted very high. Therefore, it is better to let Thievul stay in the battle to be knocked out and switch in a faster pokemon in a force switch.
**Action** $a_t$: Move: focusblast

### B.4 LAST-THOUGHTS

**Input**: observation $o_t$ (last thoughts, game knowledge are presented in **bold**)

[System] You are playing a Pokemon battle and the goal is to win. Below is an example to teach you how to reason and select an action.
===Example Start===
...
===Example End===
Here is the real case:
Historical turns:
Turn 2: Froslass used Poltergeist, which damaged opposing Cobalion's HP by 29% (30% left). Opposing Cobalion used Iron Head, which was super effective to Froslass and damaged Froslass's HP by 100% (0% left). Froslass fainted. Froslass outspeeded opposing Cobalion. You sent out Darmanitan.
Current turn:
Opponent has 6 pokemons left.
Opponent current pokemon:cobalion
Type:STEELFIGHTING,HP:30%,Atk:370(2 stage),Def:246,Spa:185,Spd:157,Spe:214
**cobalion as defender, FIGHTING,GROUND,FIRE deal 2x damage; ICE,GRASS,DARK only deal 0.5x damage; ROCK,BUG only deal 0.25x damage to cobalion**
cobalion as attacker, STEEL deal 2x damage to ICE,ROCK pokemon; STEEL deal 0.5x damage to FIRE pokemon; FIGHTING deal 2x damage to ICE,ROCK,DARK pokemon; FIGHTING deal 0.5x damage to PSYCHIC,BUG pokemon

Your current pokemon:darmanitangalar,
Type:ICE,HP:100%,Atk:263,Def:131,Spa:92,Spd:131,Spe:193
Your darmanitangalar has 4 moves can take:
flareblitz:Type:FIRE,Cate:Physical,Power:128,Acc:100%,
**Effect:User takes 1/3 the damage inflicted in recoil. Has a 10% chance to burn the target.**
uturn:Type:BUG,Cate:Physical,Power:75,Acc:100%,
**Effect:User must switch out after attacking.**
earthquake:Type:GROUND,Cate:Physical,Power:107,Acc:100%,
**Effect:Inflicts regular damage and can hit Dig users.**
iciclecrash:Type:ICE,Cate:Physical,Power:91,Acc:90%,
**Effect:Has a 30% chance to make the target flinch.**
You have 4 pokemons can switch:
crustle:Type:BUGROCK,HP:100%,Atk:222,Def:255,Spa:156,Spd:172,Spe:122,
Moves:[stoneedge,ROCK],[earthquake,GROUND],[xscissor,BUG]
grapploct:Type:FIGHTING,HP:100%,Atk:252,Def:204,Spa:170,Spd:187,Spe:121,
Moves:[icepunch,ICE],[drainpunch,FIGHTING],[suckerpunch,DARK]
audino:Type:NORMAL,HP:100%,Atk:161,Def:208,Spa:161,Spd:208,Spe:143,
Moves:[knockoff,DARK]
mesprit:Type:PSYCHIC,HP:100%,Atk:181,Def:225,Spa:225,Spd:225,Spe:183,
Moves:[psychic,PSYCHIC],[icebeam,ICE],[energyball,GRASS]
**Your thought from last step:** Cobalion has boosted its Attack by 2 stages, making it a significant threat. It is weak to FIGHTING, GROUND, and FIRE moves. Crustle has Earthquake, a GROUND move that is super effective against Cobalion, but Crustle is slower and vulnerable to Cobalion's FIGHTING moves. Grapploct has Drain Punch, a FIGHTING move that is super effective, but it is also slower and vulnerable to Cobalion's FIGHTING moves. Darmanitan-Galar has Flare Blitz, a FIRE move that is super effective, and it is faster than Cobalion, allowing it to attack first. Audino and Mesprit do not have super-effective moves and are not ideal choices. Darmanitan-Galar is the best choice due to its speed advantage and super-effective Flare Blitz.

**Output**:

**Thoughts**: Darmanitan-Galar is currently on the field and has a speed advantage over Cobalion, allowing it to attack first. Cobalion is at 30% HP and is weak to FIRE moves. Darmanitan-Galar's Flare Blitz is a FIRE move that will deal 2x damage to Cobalion. Although Flare Blitz has recoil damage, the priority is to eliminate the immediate threat posed by Cobalion's boosted Attack. Therefore, using Flare Blitz is the best option to potentially knock out Cobalion before it can attack.
**Action** $a_t$: Move: flareblitz

We can observe from the feedback described in $o_t$ that our Froslass fainted in the last turn, triggering a force switch. The LLM chose to switch in Darmanitan with the thoughts from the last step and opted to attack with Flare Blitz in this step. If the last thought is not provided, given that the opponent has already doubled its Attack stats, the LLM is very likely to behave inconsistently, as described in Appendix C.

## B.5 REFLEXION

**Phase 1: Reflection on the last step** $t-1$

**Input**: Observation $o_{t-1}$, Action $a_{t-1}$, Feedback $f_{t-1}$

[System] Make a reflection, given the state, action and its outcome in a Pokémon battle. Below is an example to teach you how to reflect on a previous battle step:
===Example Start===
...
===Example End===
Here is the real case:

Historical turns: Turn 12: opposing Sylveon used Psyshock. It was super effective to Dragalge. It damaged Dragalge's HP by 56% (0% left). Dragalge fainted. You sent out Clefable.
Current turn:
Opponent has 4 pokemons left.
Opponent's known pokemon off the field:drednaw
Opponent current pokemon:sylveon,
Type:FAIRY,HP:100%,Atk:156,Def:156,Spa:345(1 stage),Spd:394(1 stage),Spe:147
**sylveon as defender, FIGHTING,DARK,BUG only deal 0.5x damage; DRAGON have no effect to sylveon**
**sylveon as attacker, FAIRY deal 2x damage to FIGHTING,DRAGON,DARK pokemon; FAIRY deal 0.5x damage to FIRE pokemon**
Your current pokemon:clefable,
Type:FAIRY,HP:100%,Atk:119,Def:167,Spa:203,Spd:195,Spe:146 Your clefable has 4 moves can take:
moonblast:Type:FAIRY,Cate:Special,Power:73,Acc:100%,
**Effect:Has a 30% chance to lower the target's Special Attack by one stage.**
calmmind:Type:PSYCHIC,Cate:Status,Power:0,Acc:100%,
**Effect:Raises the user's Special Attack and Special Defense by one stage.**
fireblast:Type:FIRE,Cate:Special,Power:85,Acc:85%,
**Effect:Has a 10% chance to burn the target.**
softboiled:Type:NORMAL,Cate:Status,Power:0,Acc:100%,
**Effect:Heals the user by half its max HP.**
You have 4 pokemons can switch:
galvantula:Type:BUGELECTRIC,HP:19%,Atk:131,Def:146,Spa:206,Spd:146,Spe:224,
Moves:[thunder,ELECTRIC],[bugbuzz,BUG],[voltswitch,ELECTRIC]
turtonator:Type:FIREDRAGON,HP:39%,Atk:136,Def:275,Spa:201,Spd:191,Spe:109,
Moves:[fireblast,FIRE],[dracometeor,DRAGON]
malamar:Type:DARKPSYCHIC,HP:100%,Atk:193,Def:187,Spa:155,Spd:166,Spe:163,
Moves:[superpower,FIGHTING],[knockoff,DARK]
zarudedada:Type:DARKGRASS,HP:63%,Atk:232,Def:209,Spa:154,Spd:193,Spe:209,
Moves:[powerwhip,GRASS],[darkestlariat,DARK]
**Chosen Action** ($a_{t-1}$): Move: fireblast
**Outcome** ($f_{t-1}$): Clefable used Fire Blast, which damaged opposing Sylveon's HP by 16% (84% left). Opposing Sylveon outspeeded Clefable.

**Output**: Reflection

**Reflection**: The chosen action to use Fire Blast against Sylveon was not optimal. Sylveon, being a Fairy-type, does not have a weakness to Fire-type moves, and the damage dealt was minimal. Additionally, Sylveon outspeeded Clefable, allowing it to attack first and deal significant damage with Hyper Voice, reducing Clefable's HP by 65%. Given Sylveon's high Special Defense and the fact that it had a Special Attack boost, a better strategy would have been to use Moonblast, which could potentially lower Sylveon's Special Attack, or to use Calm Mind to boost Clefable's Special Defense and Special Attack, preparing for a more effective counterattack. Switching to a Pokémon with higher resistance to Fairy-type moves, such as Turtonator, could have also been considered, although its low HP would have made it a risky move.

**Phase 2: Action generation**

**Input**: Observation $o_t$, Reflection on the last step

[System] You are playing a Pokemon battle and the goal is to win. Output the best action.
Historical turns:
Turn 13: Opposing Sylveon used Hyper Voice, which damaged Clefable's HP by 65% (35% left). Clefable used Fire Blast. It damaged opposing Sylveon's HP by 16% (84% left). Opposing Sylveon outspeeded Clefable.
Current turn:

Opponent has 4 pokemons left. Opponent's known pokemon off the field:drednaw

Opponent current pokemon:sylveon,

Type:FAIRY,HP:90%,Atk:156,Def:156,Spa:345(1 stage),Spd:394(1 stage),Spe:147

**sylveon as defender, FIGHTING,DARK,BUG only deal 0.5x damage; DRAGON have no effect to sylveon**

**sylveon as attacker, FAIRY deal 2x damage to FIGHTING,DRAGON,DARK pokemon; FAIRY deal 0.5x damage to FIRE pokemon**

Your current pokemon:clefable,Type:FAIRY,HP:35%,Atk:119,Def:167,Spa:203,Spd:195,Spe:146

Your clefable has 4 moves can take:

moonblast:Type:FAIRY,Cate:Special,Power:73,Acc:100%,

**Effect:Has a 30% chance to lower the target's Special Attack by one stage.**

calmmind:Type:PSYCHIC,Cate:Status,Power:0,Acc:100%,

**Effect:Raises the user's Special Attack and Special Defense by one stage.**

fireblast:Type:FIRE,Cate:Special,Power:85,Acc:85%,

**Effect:Has a 10% chance to burn the target.**

softboiled:Type:NORMAL,Cate:Status,Power:0,Acc:100%,

**Effect:Heals the user by half its max HP.**

You have 4 pokemons can switch:

galvantula:Type:BUGELECTRIC,HP:19%,Atk:131,Def:146,Spa:206,Spd:146,Spe:224,

Moves:[thunder,ELECTRIC],[bugbuzz,BUG],[voltswitch,ELECTRIC]

turtonator:Type:FIREDRAGON,HP:39%,Atk:136,Def:275,Spa:201,Spd:191,Spe:109,

Moves:[fireblast,FIRE],[dracometeor,DRAGON]

malamar:Type:DARK&PSYCHIC,HP:100%,Atk:193,Def:187,Spa:155,Spd:166,Spe:163,

Moves:[superpower,FIGHTING],[knockoff,DARK]

zarudedada:Type:DARK&GRASS,HP:63%,Atk:232,Def:209,Spa:154,Spd:193,Spe:209,

Moves:[powerwhip,GRASS],[darkestlariat,DARK]

**Refection on the last step**: The chosen action to use Fire Blast against Sylveon was not optimal. Sylveon, being a Fairy-type, does not have a weakness to Fire-type moves, and the damage dealt was minimal. Additionally, Sylveon outspeeded Clefable, allowing it to attack first and deal significant damage with Hyper Voice, reducing Clefable's HP by 65%. Given Sylveon's high Special Defense and the fact that it had a Special Attack boost, a better strategy would have been to use Moonblast, which could potentially lower Sylveon's Special Attack, or to use Calm Mind to boost Clefable's Special Defense and Special Attack, preparing for a more effective counterattack. Switching to a Pokémon with higher resistance to Fairy-type moves, such as Turtonator, could have also been considered, although its low HP would have made it a risky move.

**Output**:

**Action** $a_t$: Switch: malamar

We can observe that the LLM is not satisfied with the action in the last step and choose to switch in current step, increasing the inconsistency.

## C  ACTION INCONSISTENCY

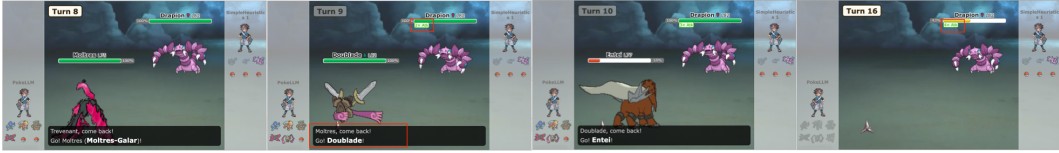

Figure 5: When facing a powerful Pokémon, the LLM switches different Pokémon in three consecutive steps to prevent its Pokémon from fainting. However, this gives the opponent three free turns to quadruple its attack stats and quickly defeat the agent's entire team.

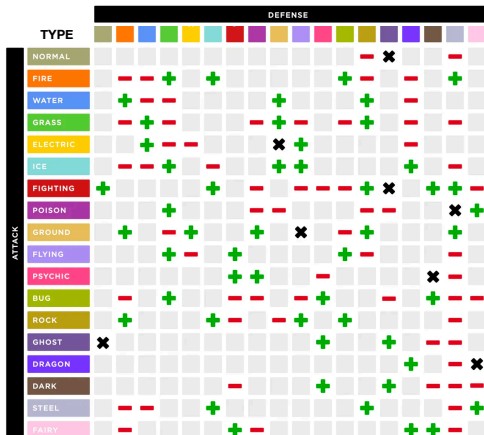

Figure 6: Type effectivness chart. "+" denotes super-effective (2x damage); "−" denotes ineffective (0.5x damage); "×" denotes no effect (0x damage). Unmarked is standard (1x) damage.

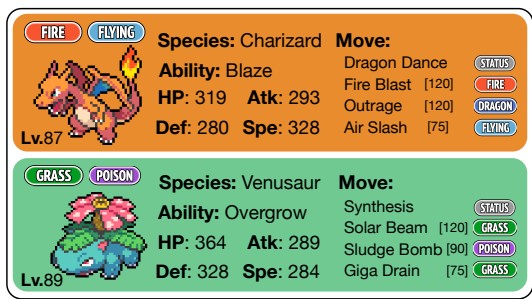

Figure 7: Two representative Pokémon: *Charizard* and *Venusaur*. Each Pokémon has type(s), ability, stats and four battle moves.

We observe that when facing a powerful Pokémon, LLMs are more tend to behave inconsistently. As illustrated in Figure 5, starting from turn 8, the agent chooses to continuously switch to different Pokémon in three consecutive turns, giving the opposing Pokémon three free turns to boost its attack stats to four times and take down the agent's entire team quickly. This phenomenon can be exacerbated by CoT but addressed by Last-Thoughts.

## D   POKÉMON KNOWLEDGE

**Species:** There are more than 1,000 Pokémon species (bul, 2024b), each with its unique ability, type(s), statistics (stats) and battle moves. Figure 7 shows two representative Pokémon: *Charizard* and *Venusaur*.

**Type:** Each Pokémon species has up to two elemental types, which determine its advantages and weaknesses. Figure 6 is the **type-effectiveness chart** that presents relationship between 18 types of attack moves and attacked Pokémon. For example, fire-type moves like "Fire Blast" of *Charizard* can cause double damage to grass-type Pokémon like *Venusaur*, while *Charizard* is vulnerable to water-type moves.

**Stats:** Stats determine how well a Pokémon performs in battles. There are four stats: (1) Hit Points (HP): determines the damage a Pokémon can take before fainting; (2) Attack (Atk): affects the strength of attack moves; (3) Defense (Def): dictates resistance against attacks; (4) Speed (Spe): determines the order of moves in battle.

**Ability:** Abilities are passive effects that can affect battles. For example, *Charizard*'s ability is "Blaze", which enhances the power of its fire-type moves when its HP is low.

**Move:** A Pokémon can learn four battle moves, categorized as attack moves or status moves. An attack move deals instant damage with a power value and accuracy, and associated with a specific type, which often correlates with the Pokémon's type but does not necessarily align; A status move does not cause instant damage but affects the battle in various ways, such as altering stats, healing or protect Pokémon, or battle conditions, *etc*. There are 919 moves in total with distinctive effect (bul, 2024a).

