# OpenReview forum: "PokéLLMon: A Grounding and Reasoning Benchmark for Large Language Models in Pokémon Battles"
_ICLR.cc/2025/Conference — ICLR 2025 Conference Withdrawn Submission_

### Official Review · Reviewer_mcz9 · 2024-10-28

**Soundness:** 2
**Presentation:** 3
**Contribution:** 2
**Rating:** 6
**Confidence:** 3

**Summary:**

This paper introduces a benchmark called PokeLLMon. The benchmark is based on adversarial gameplay of Pokemon battles, which aims at evaluating LLMs' reasoning and grounding ability. For grounding, the paper studies the effect of two methods, i.e., a retrieval-based method and a learning-based method. For reasoning, the paper also studies how different prompt-based test-time inference methods affect LLMs' reasoning performance. Lastly, they address that LLMs can be easily misdirected by human players' attrition and misdirection strategies.

**Strengths:**

Here are the strengths w.r.t to different dimensions:

Originality:
1. The paper is the first to propose a benchmark on evaluating LLMs' reasoning and grounding with the game of Pokémon.
2. The paper also finds a new perspective, i.e., action consistency, to evaluate LLMs' reasoning ability.

Quality:
1. The authors have done well-round evaluations and study w.r.t grounding and reasoning in the gameplay of Pokémon, including human evaluation.
2. For the study of grounding and reasoning, the authors have also incorporated various techniques, making their claims more convincing.

Clarity:
1. The paper is well-written and it is easy to follow the authors' ideas while reading.

Significance:
1. The paper studies an important topic, i.e., grounding and complex reasoning in adversarial gameplay for LLMs.

**Weaknesses:**

Here are some concerns of the paper:
1. The perspective of action consistency is a good point, whereas the actual implementation in the paper seems a bit superficial for me, i.e., the use of consecutive switch rate. Intuitively, a human player would not consecutively switch Pokémon, and for me it is more like a commonsense knowledge instead of something that requires reasoning ability. As is also mentioned in the paper, the LLMs may be easily biased by the prompt, which makes it more confusing whether the performance gain of the test-time inference methods are coming the boosts in LLMs' reasoning or solely coming the reduction of LLMs' biases. Apart from that, I think switch rate sounds more convincing than consecutive switch rate., and a simple sanity check can be forcing the LLMs' not to consecutively switch Pokémon.
2. The analysis in line 408-442 sounds weak and deviated to me. Instead of analyzing how those methods incorporate biases, I would prefer analysis of the actual boost/drop in reasoning ability. I think it would be good to carry out the experiments in a more constrained setting, where the biases can be mitigated.
3. The results in Table 5 are meant for showing that DPO and RFT can improve grounding. However, the table only reports win-rate and battle score, whose improvement may come from either boost in grounding or boost in reasoning. I think it'd be better to do some more studies on this, for example, carrying out knowledge examination on these tuned models.

**Questions:**

I also have some questions/suggestions for the author:

Suggestions:
1. Maybe an example of inadmissible action would be helpful for the readers that are unfamiliar with Pokémon, since in Table 3, most of the actions in $n=1$ and $n=3$ of the open-source LLMs are inadmissible.
2. typo: rejected sampling -> rejection sampling

Questions:
1. Have you tried incorporating basic rules in the system prompts? I couldn't find any in the appendix but based on my understandings, sometimes the LLMs not only lack of factual knowledge of a game but also fundamental rules/curriculums. Recent works on LLM playing text-based strategic games also follow this fashion [1,2,3]. It may be one of the most basic ways to ground LLMs to play strategic games.

```
[1] Wang, Shenzhi, et al. "Avalon's game of thoughts: Battle against deception through recursive contemplation." arXiv preprint arXiv:2310.01320 (2023).
[2] Light, Jonathan, et al. "Avalonbench: Evaluating llms playing the game of avalon." NeurIPS 2023 Foundation Models for Decision Making Workshop. 2023.
[3] Xu, Yuzhuang, et al. "Exploring large language models for communication games: An empirical study on werewolf." arXiv preprint arXiv:2309.04658 (2023).
```

---

> ### Author Response · Authors · 2024-11-19
> **Author Rebuttal (I): Reviewer mcz9**
>
> We sincerely thank the reviewer for the insightful comments. Please find our point-to-point responses below:
>
> **W1+Q1**: Consecutively switching might be due to a lack of commonsense knowledge instead of reasoning ability. **A sanity check by forcing the LLMs not to consecutively switch Pokémon** (with adding rules into the system prompt) can give some additional insights.
>
> We thank the reviewer for encouraging comments on action consistency in our benchmark and for intriguing ideas to inspire us to conduct more experiments.
>
> Indeed, in the section on knowledge grounding, we observed that the performance can be influenced by both the knowledge and the implicit reasoning capability. Previously, we did not include commonsense knowledge in the system prompt. Following your suggestion, we add additional rules below in the system prompt to constrain consecutive switching and state the negative impact of switching:
>
> ```
> “Avoid switching if you switched in the last step. If you choose to switch, you will forfeit your move for this turn, and your switch-in Pokémon will take damage from the opponent's attack. Therefore, when considering a switch, take into account the speed, type resistance, and defense of your switch-in Pokémon to ensure it can withstand the damage from the opposing Pokémon.”
> ```
>
> The comparison results between CoT and CoT w/ the new rule are given below with all the other settings remain the same:
>
> | Model        | Win Rate | Battle Score | Switch Rate | Con. Switch Rate |
> |--------------|----------|--------------|-------------|------------------|
> | CoT          | 0.3713   | 5.1270       | 0.3344      | 0.2647           |
> | CoT w/ rule  | 0.3721   | 5.0346       | 0.3495      | 0.2897           |
>
> We observe that CoT with this new rule does not show improvement compared to CoT without using this rule. We conjecture that the reason is because that the in-context example we used for prompting chain-of-thought has a similar effect in discouraging consecutive switching, as shown below:
>
> ```json
> {
>   "Thought": "This is not a force switch, therefore if I switch in a Pokemon, it will bear a free attack from the opponent. Given shiftry's speed (189) is higher than the opposing steelix (148), shiftry can outspeed steelix. And since FIRE attack deal 2x damage to steelix, I choose to use heatwave, which will lead to a significant damage, even thought it was missed in last turn.",
>   "move":"heatwave"
> }
> ```
>
> The above results demonstrate that the bias cannot be mitigated by adding constraints in either system prompt or in-context examples, which indicates the efficacy of LastThought.
>
>
> **W2**: **Analyze the actual boost/drop in reasoning ability by additional experiments in a more constrained setting**
>
> We try to disentangle the effect of reasoning in finding good strategies from mitigating the bias in another way. The choice is to violently set “consecutive switching” as inadmissible by removing the switching options if the agent just switches in the last step. With this hand-coded rule, we reimplement CoT as CoT(mute) and report the results below. We note that this rule does not truly reduce the inconsistency inherent in reasoning, but provides a way to probe the effect of reasoning.
>
> | Model | Win Rate | Battle Score | Switch Rate | Con. Switch Rate |
> |-------------------|----------|--------------|-------------|------------------|
> | IOPrompt | 0.4217 | 5.413 | 0.3356 | 0.2442 |
> | IOPrompt (mute) | 0.4082 | 5.393 | 0.2719 | 0.0 |
> | CoT | 0.3713 | 5.127 | 0.3344 | 0.2647 |
> | CoT (mute) | 0.4459 | 5.738 | 0.2594 | 0.0 |
> | LastThoughts | 0.4667 | 5.840 | 0.2227 | 0.0861 |
>
>
> We make two observations from the results:
>
> (1) The original result of CoT is indeed a composite effect of reasoning. By mitigating the bias caused by consecutive switching, we disentangle the effect of finding better strategies, achieving a +2.42% improvement compared to IOPrompt, and a +7.46% improvement compared to CoT in win rate, showing that reasoning is effective in our benchmark when the bias is mitigated. In contrast, IOPrompt (mute) does not show improvements, suggesting that the bias primarily stems from explicit reasoning;
>
> (2) LastThoughts still outperforms CoT (mute) by +2.08% in win rate, indicating that its effectiveness stems not only from mitigating bias but also from improved reasoning. Although we cannot further pinpoint which specific factor the improvements stem from—whether they are due to finding better strategies or reducing inconsistencies represented in other ways—we believe that effective reasoning is inherently multifaceted, i.e., it needs to find good strategies consistently.

---

> ### Author Response · Authors · 2024-11-19
> **Author Rebuttal (II): Reviewer mcz9**
>
> **W3:** Table 5 only reports win-rate and battle score of DPO and RFT for improrving grounding. **Carry out knowledge examination on these tuned models** to investigate where the improvement comes from.
>
> Thank you for another insightful suggestion! We conduct additional experimentation to examine whether game knowledge can be learned from self-play. As trained LLMs are not aligned for chatting, we adopt linear probing approach for knowledge examination: Specifically, we use fine-tuned LLMs after self-play to extract the representation of the below questions and then train a logistic regression model on 50% of examples and test on the left examples:
>
> ```
> Question: In Pokémon battles, a {TYPE1} attack is ____ against a {TYPE2} Pokémon?\nA. Super Effective (2x Damage)\nB. Standard (1x Damage)\nC. Not Effective (0.5x Damage)\nD. Zero Effect (0x Damage)\nOutput:
> ```
> Label $y \in$ {A, B, C, D}
>
> We report the F1 score of each category and weighted F1 in below:
>
> | Model  | F1 (B)  | F1 (A) | F1 (C) | F1 (D) | F1    | Win Rate|  Battle Score |
> |--------|---------|--------|--------|--------|--------| ------------|------------|
> | Origin | 0.7103  | 0.1200 | 0.2642 | 0.0000 | 0.5231 | 0.1075  | 3.904 |
> | RFT    | 0.7589  | 0.1000 | 0.2593 | 0.0000 | 0.5510 | 0.1161  | 4.194 |
> | DPO    | 0.7477  | 0.1860 | 0.2308 | 0.0000 | 0.5533 | 0.1212 | 4.207 |
>
>
> Observation: After grounding with self-play experience, LLMs show improved performance on the type-effectiveness prediction task, suggesting that LLMs indeed learn game knowledge from self-play. As the current grounding does not involve explicit reasoning, we believe that the improvement in gameplay primarily results from the learning of game knowledge. We leave the training of explicit reasoning from self-play as our future work.

---

> ### Author Response · Authors · 2024-11-19
> **Author Rebuttal (III): Reviewer mcz9**
>
> **Suggestion 1**: Provide an example of inadmissible action
>
> Below is an example for an inadmissible output of open-sourced LLMs. The input includes an in-context example (n=1) for guiding open-sourced LLM output action in each step. However, the LLM repeats the output in the 1-shot example instead of selecting an admissible action from the action candidates (4 moves and 2 switch-in Pokémon), rendering the output action inadmissible, which cannot be solved by adding the system prompt “Do not repeat the output of the example”.
>
> ```
> **Input:**
> [System] You are playing a Pokemon battle and the goal is to win. Select the best action and
> output. Below is an example to teach you how to select an action. Do not repeat the output of the example.
> ===Example Start===
> Opponent has 6 pokemons left.
> Opponent current pokemon:mandibuzz:Type:DARK&FLYING,HP:100%,Atk:154,Def:219,Spa:137,Spd:203,Spe:178
> Your current pokemon:magmortar,Type:FIRE,HP:100%,Atk:170,Def:166,Spa:267,Spd:215,Spe:194
> Your magmortar has 4 moves can take:
> taunt:Type:DARK,Cate:Status,Power:0,Acc:100%
> focusblast:Type:FIGHTING,Cate:Special,Power:158,Acc:70%
> thunderbolt:Type:ELECTRIC,Cate:Special,Power:118,Acc:100%
> fireblast:Type:FIRE,Cate:Special,Power:145,Acc:85%
> You have 5 pokemons can switch:
> golduck:Type:WATER,HP:100%,Atk:146,Def:183,Spa:213,Spd:187,Spe:195,Moves:[focusblast,FIGHTING],[scald,WATER]
> corviknight:Type:FLYING&STEEL,HP:100%,Atk:181,Def:209,Spa:128,Spd:178,Spe:150,Moves:[bravebird,FLYING],[bodypress,FIGHTING]
> gigalith:Type:ROCK,HP:100%,Atk:269,Def:260,Spa:146,Spd:178,Spe:88,Moves:[superpower,FIGHTING],[earthquake,GROUND],[stoneedge,ROCK]
> drampa:Type:NORMAL&DRAGON,HP:100%,Atk:108,Def:195,Spa:281,Spd:206,Spe:111,Moves:[dracometeor,DRAGON],[hypervoice,NORMAL],[fireblast,FIRE]
> dusknoir:Type:GHOST,HP:100%,Atk:224,Def:285,Spa:163,Spd:285,Spe:128,Moves:[shadowsneak,GHOST],[earthquake,GROUND],[icepunch,ICE],[poltergeist,GHOST]
> Output:{"move":"thunderbolt"}
> ===Example Ends===
> Here is the real case:
> Historical turns:
> Turn 12: Bouffalant used Megahorn, which was super effective to opposing Reuniclus. It damaged opposing Reuniclus's HP by 59% (41% left). Opposing Reuniclus used Focus Blast, which was super effective to Bouffalant. It damaged Bouffalant's HP by 53% (0% left). Bouffalant fainted. Bouffalant outspeeded opposing Reuniclus. You sent out Doublade.
> Current turn:
> Opponent has 2 pokemons left.
> Opponent current pokemon:reuniclus,
> Type:PSYCHIC,HP:41%,Atk:157,Def:174,Spa:258,Spd:191,Spe:99
> Your current pokemon:doublade,
> Type:STEEL&GHOST,HP:10%,Atk:228,Def:293,Spa:121,Spd:128,Spe:105
> Your doublade has 4 moves can take:
> swordsdance:Type:NORMAL,Cate:Status,Power:0,Acc:100%,
> ronhead:Type:STEEL,Cate:Physical,Power:105,Acc:100%,
> closecombat:Type:FIGHTING,Cate:Physical,Power:157,Acc:100%,
> shadowsneak:Type:GHOST,Cate:Physical,Power:52,Acc:100%,
> You have 2 pokemons can switch:
> azelf:Type:PSYCHIC,HP:0%,Atk:249,Def:160,Spa:249,Spd:160,Spe:233,
> Moves:[uturn,BUG],[psychic,PSYCHIC],[fireblast,FIRE]
> tsareena:Type:GRASS,HP:100%,Atk:250,Def:213,Spa:132,Spd:213,Spe:169,
> Moves:[rapidspin,NORMAL],[tripleaxel,ICE],[powerwhip,GRASS]
>
> **Output**: {"move":"thunderbolt"}
> ```
>
>
> **Suggestion 2:** For typos, we will revise them in the updated version.
>
> **Q1**: Basic rules in the systems prompts
>
> Following your suggestions, we have provided a sanity check in the previous response (W1+Q1). We thank the reviewer for sharing three related papers on system prompt design to incorporate commonsense game knowledge. We will include these papers in the related work section of our updated version, along with relevant discussions.
>
> ----
> We sincerely thank this reviewer for the insightful comments and suggestions. If the reviewer finds our responses satisfactory, we hope that you would consider raising your scores.

---

> > ### Comment · Reviewer_mcz9 · 2024-11-24
> >
> > Thank you for your clarification and additional experiments. I will raise my score from 5 to 6 since you've addressed all of my concerns.

---

> ### Author Response · Authors · 2024-11-24
> **Thanks for your response**
>
> Dear Reviewer mcz9,
>
> We truly appreciate your encouraging and insightful comments and suggestions! Thank YOU.

---

### Official Review · Reviewer_Tc95 · 2024-11-01

**Soundness:** 3
**Presentation:** 3
**Contribution:** 2
**Rating:** 5
**Confidence:** 4

**Summary:**

The paper introduces a new benchmark aimed at enhancing the grounding and reasoning capabilities of Large Language Models (LLMs) using the context of Pokémon battles. This benchmark, PokéLLMon, is designed to challenge LLMs with complex, dynamic, and strategic gameplay that goes beyond current models' capabilities by integrating rich, fictional game knowledge.

**Strengths:**

1. This application to a unique domain (Pokémon battles) itself is novel, as it combines elements of strategic game play with AI training in a way that is not commonly explored in LLM research.
2. The paper is underpinned by robust experimental setups, including comparisons with popular environments in LLM research, demonstrating the benchmark’s effectiveness in identifying gaps in current LLM capabilities regarding game knowledge and strategic reasoning.
3. The paper is well-structured, with a clear exposition of the problem area, the novelty of the PokéLLMon environment, and the methods used for evaluation.

**Weaknesses:**

1. The specificity of the Pokémon battle environment might limit the generalizability of the findings. While the paper demonstrates significant advancements within this particular domain, it may not be clear how these findings extend to other strategic or adversarial settings outside of gaming.
2. The requirement for specific knowledge to effectively utilize the PokéLLMon benchmark could deter its use as a standard testing ground compared to more accessible or universally relevant environments.

**Questions:**

1. While the paper provides empirical evaluations, there could be a need for deeper analysis regarding the failure modes of LLMs within this benchmark. Understanding why certain models fail or succeed under specific conditions can offer more nuanced insights into the models’ reasoning and strategic capabilities.
2. In the paper, 'In intense adversarial games, action consistency is an important indicator of performance.' and you give the reason 'LLM frequently switches Pokemon in consecutive steps and wastes chances to attack.' So I think action consistency seems to be narrow as a metric. And it could be better if you give more explanations or experiments to demonstrate its importance beyond the specific gaming task. Or I will take the comments that LastThoughts only conservatively imitates the previous steps to avoid wastes and is limited in LLM reasoning.

---

> ### Author Response · Authors · 2024-11-19
> **Author Response (I): Reviewer Tc95**
>
> We thank the reviewer for the constructive and insightful comments. We make our best efforts to address each of the comments below:
>
> **W1 + Q2**: **Can specificity of Pokémon battle environment (e.g., action consistency) be extended to other strategic settings outside gaming?**
> Yes. It can. We below discuss this potential from two perspectives common to many decision-making environments.
>
> **Action consistency** is defined as the degree to which an agent acts consistently and coherently with its past experiences [1]. It is a general concept in interactive environments but measured using specific metrics in our benchmark. To illustrate its generality, we first explain the inconsistency phenomenon common to other strategic environments beyond Pokémon battles:
>
> (1)	Inconsistency over time-steps:
> In generative agents for human behavior simulation [2], inconsistency refers to the phenomenon where agents repeat the same actions multiple times even the action has already been done, e.g., eating lunch at 12pm, then again at 12:30 pm. at 1pm, despite knowing that it has already eaten lunch twice. This phenomenon is similar to consecutive switching in our benchmark. In comparison, [2] avoids inconsistency in a different way: each agent has a static plan for the entire day (e.g., 12:00 PM: eat lunch, 12:30-1:30 PM: take a break, etc.). At specified time points, the agent is prompt with the static plan to improve its action consistency over time. This approach is not applicable to our benchmark as Pokémon battles are highly dynamic and hard to foresee via static planning.
>
> (2)	Inconsistency due to the stochasticity of reasoning:
> Chain-of-Thoughts (CoT) generates intermediate tokens to form a reasoning path toward the final answer. It is stochastic and influenced by the generation probability distribution and the sampling process of each token. One-time reasoning is a specific sample from a vast number of possible reasoning paths and is likely not to lead to a consistent answer [3]. A method to find a consistent answer (SC[3]) is by conducting majority voting on multiple CoT reasoning paths, which has demonstrated improvements in various reasoning tasks, including arithmetic reasoning (GMS8K, SVAMP) and common sense reasoning (CommonsenseQA, ARC), suggesting the general efficacy of enhancing consistency.
>
> **Experimental evidence**: Following your suggestions, we conducted experiments to verify the efficacy of enhancing consistency beyond our benchmark. Experiments are conducted on ScienceWorld [4], a popular text-based environment that asks the agent to fulfill tasks like doing science experiments. We implement CoT over GPT-4o to conduct new reasoning at every single step, and Last-thoughts that refine the previous k thoughts for reasoning (k=1,2). The task template and few-shot examples are borrowed from [5]. Below we report the average reward on the testing set.
>
> | Method            | Win Rate |
> |-------------------|----------|
> | CoT               | 0.7011   |
> | Last-Thoughts (1) | 0.7116   |
> | Last-Thoughts (2) | 0.6798   |
>
> We observe that integrating one last thought increases the average reward, while introducing more previous thoughts decreases performance, as the previous thoughts become outdated in the new step. This suggests that striking a balance between consistency and dynamism is essential in interactive environments.
>
> Overall, action consistency impacts the performance, but directly measuring  action consistency quantitatively is challenging in existing benchmarks. In comparison, our benchmark can measure the action consistency using the (consecutive) switch rate, and its adversarial nature makes the cost of generating inconsistent actions much heavier and more evident, compared to conventional non-adversarial environments (ALFWorld, ScienceWorld, WebShop, etc.)
>
> [1] Xu, Xinrun, et al. "A survey on game playing agents and large models: Methods, applications, and challenges." arXiv preprint arXiv:2403.10249 (2024).
>
> [2] Park, Joon Sung, et al. "Generative agents: Interactive simulacra of human behavior." Proceedings of the 36th annual acm symposium on user interface software and technology. 2023.
>
> [3] Wang, Xuezhi, et al. "Self-consistency improves chain of thought reasoning in language models." arXiv preprint arXiv:2203.11171 (2022).
>
> [4] Wang, Ruoyao, et al. "ScienceWorld: Is your Agent Smarter than a 5th Grader?." Proceedings of the 2022 Conference on Empirical Methods in Natural Language Processing. 2022.
>
> [5] Song, Yifan, et al. "Trial and error: Exploration-based trajectory optimization for llm agents." arXiv preprint arXiv:2403.02502 (2024).

---

> ### Author Response · Authors · 2024-11-19
> **Author Rebuttal (II): Reviewer Tc95**
>
> **W2: Would the use of Pokemon-specific knowledge deter the benchmark utility compared to more accessible environments?**
>
> Our answer is No. We believe that the existence of game knowledge and strategic game-play makes PokéLLMon unique for testing the potential of LLMs in reasoning and grounding capability. Below, we discuss our argument from three aspects:
>
> (1) For testing-time approaches, PokéLLMon offers well-structured game knowledge as a configurable option for generating observation prompts. Researchers do not need to expend effort in collecting game knowledge to use our benchmark as a testbed.
>
> (2) For training-time methods, external game knowledge is **not** required, as LLMs can learn game knowledge through self-play, similar to how AlphaGo Zero learns the rules of playing Go game and OpenAI Five learns to play DOTA. The strategic gameplay makes the environment more challenging, thereby fostering more advanced techniques.
>
>
> To illustrate this, we provide a knowledge probing experiment to test whether game knowledge can be learned from self-play:
>
> Specifically, we use LLMs after grounding with self-play to extract the representation of the following questions (derived from the type-effectiveness chart in Table 6 of our appendix), and train a logistic regression model on 50% of examples and test on the left examples:
>
> ```
> Question: In Pokémon battles, a {TYPE1} attack is ____ against a {TYPE2} Pokémon?\nA. Super Effective (2x Damage)\nB. Standard (1x Damage)\nC. Not Effective (0.5x Damage)\nD. Zero Effect (0x Damage)\nOutput:
> ```
> Label $y \in$ {A, B, C, D}
>
> We below report the F1 score of each category and weighted F1 before and after self-play:
>
> | Model  | F1 (B)  | F1 (A) | F1 (C) | F1 (D) | F1    | Win Rate|  Battle Score |
> |--------|---------|--------|--------|--------|--------| ------------|------------|
> | Origin | 0.7103  | 0.1200 | 0.2642 | 0.0000 | 0.5231 | 0.1075  | 3.904 |
> | RFT    | 0.7589  | 0.1000 | 0.2593 | 0.0000 | 0.5510 | 0.1161  | 4.194 |
> | DPO    | 0.7477  | 0.1860 | 0.2308 | 0.0000 | 0.5533 | 0.1212 | 4.207 |
>
> We observe that LLMs show better performance on the type-effectiveness prediction task, showing that LLMs effectively learn game knowledge from self-play **without relying** on external knowledge. In this case, the strategic knowledge becomes an advantage since grounding becomes more challenging compared to other benchmarks.
>
> (3) Further, our benchmark features some additional advantages: The gameplay is highly dynamic and strategic with a vast number of possible combinations, and its adversarial nature provides a very high ceiling for reasoning, particularly against powerful opponents such as human experts. These features are lacking in existing benchmarks.
>
>
> **Q1: Provide a deeper analysis regarding the failure modes of LLMs**
>
> One LLM failure example for reasoning is to frequent consecutive switching, leading to consecutive switching and wasting opportunities to attack the opponent. To gain a deeper understanding of whether such a failure of reasoning is due to the inconsistencies or due to the inability to find better strategies (limited in reasoning), we conducted additional experiments on the effect of reasoning in a controlled setting:
>
> We set 'consecutive switching' as inadmissible by removing the switching options if the agent just switched in the last step. With this hand-coded rule, we reimplemented CoT as CoT(mute) and report the results obtained:
>
> | Model | Win Rate | Battle Score | Switch Rate | Con. Switch Rate |
> |-------------------|----------|--------------|-------------|------------------|
> | IOPrompt | 0.4217 | 5.413 | 0.3356 | 0.2442 |
> | IOPrompt (mute) | 0.4082 | 5.393 | 0.2719 | 0.0 |
> | CoT | 0.3713 | 5.127 | 0.3344 | 0.2647 |
> | CoT (mute) | 0.4459 | 5.738 | 0.2594 | 0.0 |
> | LastThoughts | 0.4667 | 5.840 | 0.2227 | 0.0861 |
>
> We make two observations from the results:
>
> (1) CoT (mute) outperforms IOPrompt (no explicit reasoning), suggesting that explicit reasoning can lead to better answer than directly outputting answers. This indicates that the poor performance of LLM reasoning (CoT) is primarily due to the frequent consecutive switching introduced by inconsistency in reasoning, giving us a deeper understanding of why LLM reasoning fails. In contrast, IOPrompt (mute) does not show improvements, suggesting that the inconsistency primarily stems from explicit reasoning.
>
> (2) LastThoughts still outperforms CoT (mute) by +2.08% in win rate, which suggests that the effectiveness of LastThoughts stems not only from reducing wasted attack chances but also from improved reasoning. This observation indicates that LastThoughts is not “to conservatively imitate the previous steps to avoid wastes", and is not "limited in reasoning”.
>
> ----
> We sincerely thank the reviewer for your insightful comments and suggestions. We hope you would consider to raise your scores if you finds our response satisfactory.

---

> > ### Comment · Reviewer_Tc95 · 2024-11-25
> >
> > Thank you for your reply. I will maintain my score since majority of my concern is addressed.

---

> > > ### Author Response · Authors · 2024-11-25
> > >
> > > Dear Reviewer Tc95,
> > >
> > > We are pleased to hear that our responses have addressed your concerns. Given that the rating currently stands at 5, marginally below the acceptance threshold, we kindly ask if you would consider raising your rating based on the explanations and experiments we have conducted to address your feedback.
> > >
> > > Regards,
> > > Authors

---

### Official Review · Reviewer_Loeg · 2024-11-03

**Soundness:** 2
**Presentation:** 2
**Contribution:** 1
**Rating:** 3
**Confidence:** 5

**Summary:**

The paper introduces **PokéLLMon**, a new benchmark designed to evaluate the grounding and reasoning capabilities of Large Language Models (LLMs) in the context of Pokémon battles. The authors propose grounding techniques that leverage game knowledge and self-play experience, and they analyze reasoning methods from the perspective of action consistency.

**Strengths:**

1. **Grounding Techniques**: The exploration of grounding techniques that incorporate game knowledge and self-play experience is insightful and adds value to the research.
2. **Action Consistency**: Analyzing action consistency and introducing techniques to improve it are valuable contributions to the field of interactive LLMs.

**Weaknesses:**

1. **Code Accessibility**: The provided code link is not accessible, hindering the experiments' reproducibility and the validation of the results.
2. **Lack of Comparison**: The paper does not mention or compare with a similar work available on arXiv, titled "PokéLLMon: A Human-Parity Agent for Pokemon Battles with Large Language Models" (https://arxiv.org/abs/2402.01118).
3. **Contribution Level**: The paper's contributions are relatively low for a top-tier machine learning conference. The work seems more suitable for workshops or tracks focused on datasets and benchmarks.

**Questions:**

1. The paper should include a comparison with the similar work mentioned in the review (https://arxiv.org/abs/2402.01118).
2. The author needs to discuss the generalizability of the method in this paper to other decision-making tasks.

---

> ### Author Response · Authors · 2024-11-19
> **Author Rebuttal to comments by Reviewer Loeg**
>
> **W1: Code Accessibility** Thank you. Upon receiving the review, we checked why our code link is not accessible, and learned that the AnonymousGitHub website has set an expiration date for the links, and our PokeLLMon link expired on Oct. 27. We have reset the link: https://anonymous.4open.science/r/PokeLLMon.
>
> **W2+Q1: Should provide a comparison to an ArXiv paper https://arxiv.org/abs/2402.01118** Thank you. This arXiv preprint is an earlier report of the PokeLLMon project. To obey the anonymous submission requirement, we omit this arXiv report in our ICLR submission. Further, our current ICLR submission enriches PokéLLMon as a new benchmark for grounding and reasoning techniques, incorporating more representative LLMs and approaches, such as DPO and RFT, along with PPO suggested by Reviewer ejmw.
>
> **W3: More suitable for venues on datasets and benchmarks**  As you suggested, we indeed submitted this paper to the "Datasets and Benchmarks" track of ICLR 25. The paper introduces a new benchmark for LLM grounding and reasoning, and presents new findings on existing techniques within our benchmark, supported by comprehensive experiments summarized as follows:
>
> (1) We evaluate ten representative LLMs in terms of game knowledge and gameplay performance, revealing that a lack of game knowledge causes ungrounded LLMs to struggle in gameplay;
> (2) For grounding, we highlight the importance of game knowledge and assess self-evolution techniques that enable models to learn directly from interactions with the environment;
> (3) For reasoning, we analyze existing reasoning approaches from a new perspective—action consistency—showing that explicit reasoning can mitigate inconsistencies in action, while recursively integrating previous reasoning enhances consistency;
> (4) Online battles with humans suggest that even the most advanced LLMs struggle with human-level strategies, such as the attrition strategy, highlighting higher-level reasoning challenges for future research.
>
> **Novelty of PokéLLMon as a LLM benchmark**:
> (a) It features fictional game knowledge beyond the scope of existing LLMs, making it well-suited for developing grounding techniques;
> (b) The gameplay is highly dynamic and strategic, with a vast number of possible combinations and an adversarial nature, providing a new challenge for LLM reasoning, particularly against powerful opponents such as human experts.
>
> **Q2: Generalizability of action consistency to other decision-making tasks**
> Action consistency is a general concept [1] in interactive environments for improving the quality of human behavior simulation [2] and reasoning effectiveness [3]. Reasoning can introduce inconsistency due to its inherent stochasticity, e.g., Chain-of-Thoughts (CoT) as a stochastic process can generate intermediate tokens to form a reasoning path toward the final answer but the entire process is influenced by the generation probability distribution and the sampling process of each token, making one-time reasoning a specific sample from a vast number of possible reasoning paths and is likely not to lead to a consistent answer [3].
>
> We conduct new experiments on ScienceWorld [4], a popular decision-making environment that asks the agent to fulfill tasks like doing science experiments. We implement CoT with GPT-4o to conduct new reasoning at every single step, and Last-thoughts that refine the previous k thoughts for reasoning (k=1,2). Below is the results of average reward on the testing set.
>
> | Method            | Win Rate |
> |-------------------|----------|
> | CoT               | 0.7011   |
> | Last-Thoughts (1) | 0.7116   |
> | Last-Thoughts (2) | 0.6798   |
>
>
> We observe that integrating one last thought increases the average reward with consistency enhanced, while introducing more previous thoughts decreases performance, as the they become outdated in the new step, suggesting that striking a balance between consistency and dynamism is essential in interactive environments.
>
> Overall, directly measuring action consistency quantitatively is known to be difficult in existing benchmarks. Our benchmark can  measure action consistency with (consecutive) switching rate and the cost for generating inconsistent actions in adversarial games.
>
> [1] Xu, et al. survey on game playing agents and large models: Methods, applications, and challenges. arXiv:2403.10249 (2024).
>
> [2] Park, et al. Generative agents: Interactive simulacra of human behavior." Annual acm symposium on user interface software and technology. 2023.
>
> [3] Wang, et al. Self-consistency improves chain of thought reasoning in language models. arXiv:2203.11171 (2022).
>
> [4] Wang, et al. ScienceWorld: Is your Agent Smarter than a 5th Grader?. EMNLP2022.
>
> ---
> We sincerely thank the reviewer for your comments and suggestions. If the reviewer finds our responses satisfactory, we hope that you would consider raising your scores.

---

> ### Author Response · Authors · 2024-12-01
> **Warm Reminder**
>
> Dear Reviewer Loeg,
>
> As the discussion period approaches its end, we would like to know if our response addressed your concerns?
>
> Best regards,
> Authors

---

### Official Review · Reviewer_eJmw · 2024-11-04

**Soundness:** 3
**Presentation:** 4
**Contribution:** 3
**Rating:** 8
**Confidence:** 4

**Summary:**

The paper introduces PokéLLMon, a new benchmark aimed at testing the grounding and strategic reasoning abilities of large language models (LLMs) within the complex and dynamic setting of Pokémon battles. This benchmark addresses two critical requirements for interactive environments: (1) the need for knowledge beyond what current LLMs contain and (2) tasks that require nuanced strategic reasoning. The benchmark is structured to expose LLMs to unfamiliar knowledge (such as Pokémon type effectiveness and move interactions) and to test their ability to reason and make decisions in adversarial gameplay.

**Strengths:**

1. The benchmark on Pokémon battles is nice. Those battles are complex even for human to complete. Adding Pokémon in a plus in the LLM benchmark for reasoning and game playing.
2. Clear analysis on evaluation including game knowledge examination, and game performance evaluation.
3. Comprehensive analysis on Reasoning, which is the focus on this benchmark, and introduce "Last Thought" to improve consistency.

**Weaknesses:**

1. The citation format makes it a little bit hard to read the whole paper. Would you mind changing that? Maybe like:
In other words, LLMs lack experiential grounding Mahowald et al. (2024); --> In other words, LLMs lack experiential grounding (Mahowald et al.,2024);
2. In section 5.2, since RFT and DPO are included, it's natural to consider the combination of PPO (Schulman et al., 2017) with a trained  reward model.

**Questions:**

1. For some reason, the link provided in the abstract says "expired".
2. Is it possible to include a variation of Voyager (Wang et al., 2023) in the section of reasoning? It would be too hard to construct a skill library for all the Pokémons. Maybe only for one or two Generation 1 Pokémon, like Pikachu and Squirtle? Both of them would have "Tackle", but Squirtle has "Water Gun" and Pikachu has "Thunder Shock".

---

> ### Author Response · Authors · 2024-11-20
> **Author Rebuttal (I)**
>
> We sincerely thank the reviewer for the positive feedback on our benchmark. Please find our point-to-point answers below:
>
> **W1: Citation format**
>
> Thank you for your feedback, we have revised the citation format in the updated version.
>
> **W2: Implementation of PPO [1] with a value model**
>
> Following your suggestions, we further implement LLM-PPO for LLaMA2-7b and Symbolic-PPO, a DNN (approximately 1M parameters) that takes symbolic input of observable game state. Training follows the same setting as grounding methods in the paper. The preliminary results are reported as below (the opponent is ExpertSystem):
>
> | Method        | Win Rate | Battle Score |
> |---------------|----------|--------------|
> | LLM-Origin        | 0.1075   | 3.904        |
> | LLM-DPO           | 0.1212   | 4.207        |
> | LLM-PPO       | 0.1274   | 4.389        |
> | Symbolic-PPO    | 0.0749   | 3.341        |
>
> We observe that LLM-PPO demonstrates better performance than DPO. We conjecture that this is because DPO learns to weigh each action based on the win/lose signals of the entire trajectory, which is likely contain noise and be less efficient for learning. In contrast, the value model in PPO can efficiently attribute credit to each action, thereby making the learning process more effective. By comparing LLM-PPO and Symbolic-PPO, we observe that an LLM, which understands language and has superior model capacity, performs better than a DNN.
>
>
> [1] Schulman, John, et al. "Proximal policy optimization algorithms." arXiv preprint arXiv:1707.06347 (2017).
>
> **Q1: Expiration of anonymous code link**
>
> Thank you for pointing out this. Previously we didn't notice that AnonymousGitHub website has an expiration date for links, and our link expired on Oct. 27. We have reset it for your review: https://anonymous.4open.science/r/PokeLLMon.
>
> **Q2: Extending skill library in Voyager to Pokémon moves**
>
> In Voyager (Wang et al., 2023), a skill library refers to a queryable codebase for reusing past skills in the future. Similarly, the game knowledge in our benchmark includes a move library that describes the effects of each move and can be queried during battles. Below is an example illustrating the utility of this game knowledge library:
>
>     Toxic: Toxic poisons the target, causing it to lose progressively increasing HP each turn.
>
> With this move library (knowledge), we observe that the LLM employs strategies involving multiple moves, such as an attrition strategy shown in Figure 2: the agent first uses Toxic to poison the opponent, inflicting additional poisoning damage on each turn. Then, it prolongs the battle by frequently healing itself with Recover, a move that restores 50% of its HP. As a result, the opponent gradually weakens from the poisoning damage and faints after 7 turns.
>
> ----
>
> We sincerely thank the reviewer for your inspiring and constructive review comments.

---

> > ### Comment · Reviewer_eJmw · 2024-11-23
> >
> > Thank you for your reply. I will maintain my score since majority of my concern is addressed.

---

> > > ### Author Response · Authors · 2024-11-23
> > > **Thanks for your response**
> > >
> > > Hi Reviewer eJmw,
> > >
> > > Thank you for letting us know that our response addressed your concerns. We sincerely appreciate your positive comments and valuable suggestions. Please do not hesitate to share any further ideas as we interact with other reviewers. Thanks again!
> > >
> > > Best regards,
> > > Authors

---

### Official Review · Reviewer_H5C9 · 2024-11-04

**Soundness:** 3
**Presentation:** 2
**Contribution:** 2
**Rating:** 3
**Confidence:** 4

**Summary:**

This paper introduces a benchmark, PokeLLMon, designed to assess the reasoning capabilities of large language models in interactive environments requiring extensive game-specific knowledge and strategic reasoning. Existing LLM environments tend to be limited by their reliance on commonsense knowledge, so PokeLLMon addresses this by incorporating the complex and dynamic gameplay of Pokémon battles, which require deep understanding of fictional knowledge not commonly found in LLMs. The authors empirically demonstrate that most LLMs, including open-source models, struggle in the Pokémon battle domain due to limited game knowledge and action consistency issues, which makes their benchmark challenging to current LLMs.

**Strengths:**

One of the main strengths of this paper is its comprehensive evaluation approach. The authors assess LLM performance across multiple dimensions, including game knowledge understanding, action consistency, and strategic reasoning. By using a variety of metrics—such as win rate, battle score, and newly introduced action consistency metrics like switch rate and consecutive switch rate—they provide a detailed analysis of how well models can handle the complex demands of Pokémon battles. Additionally, the evaluation includes comparisons against expert systems and human players, highlighting the models’ strengths and weaknesses in both controlled and real-world scenarios. This thorough approach offers valuable insights into the current limitations of LLMs and may encourage further research.

**Weaknesses:**

- While the paper thoroughly evaluates existing LLMs with some modifications, it does not propose or implement a novel baseline that could help inspire further research. This limits the contribution of this work.
- The presentation of some figures in this paper is not clear. In Fig. 1, Fig. 2 and Fig. 4, the text is barely readable.
- Appendix B is empty with only a title.

**Questions:**

1. In Table 3, there are human players involved in the experiments, but they are "randomly-matched human players from game server". Do you think this result can be biased? I question the reliability of this result because different players may hold very different performance, and the server may match different players for you according to your recent performance. For example, the server may match your opponent according to your Elo score, so the level of your opponent is always close to your own level.
2. What is the process to generate multiple-choice questions for game knowledge examination? And what is the scale of this question dataset?

---

> ### Author Response · Authors · 2024-11-20
> **Author Response (I): Categorization of existing baselines and addition of two new training-time baselines**
>
> We sincerely thank the reviewer for the insightful comments. We make our best efforts to address each of the comments below:
>
> **Response to W1**: Our main paper includes ten representative LLMs (Tables 2 & 3), three grounding approaches (Tables 4 & 5), and five reasoning methods (Table 6), which can be categorized as testing-time and training-time baselines.
>
> **Testing-time baselines**: Testing-time approaches fix the parameters of LLMs and primarily modify the input prompts to obtain the desired outputs. The evaluation of existing LLMs, along with five reasoning approaches (CoT, SC, ToT, Reflexion, LastThoughts), and testing-time grounding with game knowledge, can all be classified into this category. The reviewer’s comment about "LLMs with some modifications" primarily fall into this category of testing-time baselines.
>
> The testing-time baselines are designed/implemented with the purpose of showcasing unique advantages of the benchmark. For example, (1) Grounding with knowledge suggests that the benchmark requires fictional game knowledge, making training-time grounding more challenging compared to existing benchmarks; (2) Reasoning approaches are evaluated to demonstrate the importance of enhancing the action consistency in intense, adversarial games, as opposed to less dynamic environments.
>
> **Training-time baselines**: The two training-time grounding approaches, DPO[1] and RFT[2], are implemented in a cyclical two-stage process: collecting trajectories through self-play and fine-tuning using self-evolution algorithms. We observe that LLMs evolve through self-play rather than relying on external knowledge.
>
> Based on your comments, we add two additional training-time baselines and experiments to strengthen our findings: two new reinforcement learning baselines (LLM-PPO and Symbolic-PPO) are implemented for training-time grounding, both trained using the Proximal Policy Optimization (PPO) algorithm [3]. LLM-PPO adopts a LLAMA-2-7B model as the policy and DNN-PPO is a 1M-parameter neural network that processes a symbolic input of the game observable state. We also introduced a variant of LLM-PPO with all parameters randomly initialized (LLM-PPO-rand). Below, we report the gameplay performance of baselines following the same settings as in our paper (Origin denotes no self-play and the opponent is ExpertSystem):
>
> | Method               | Win Rate | Battle Score |
> |--------------------|------------|--------------|
> | LLM-Origin        | 0.1075     | 3.904        |
> | LLM-DPO          | 0.1212     | 4.207        |
> | LLM-PPO          | 0.1274   |4.389      |
> | LLM-PPO-rand  | 0.0381    | 2.791        |
> | DNN-PPO   | 0.0749    | 3.341        |
>
> We observe that LLM-PPO demonstrates better performance than DPO for two reasons: (i) DPO learns to weigh each action based on the win/lose signals of the entire trajectory, which is likely contain noise and be less efficient for learning. (ii) The value model in PPO can efficiently attribute credit to each action, thereby making the learning process more effective.
>
> By comparing LLM-PPO and Symbol-PPO, we observe that an LLM, which understands language and has superior model capacity, performs better than a DNN (Symbolic-PPO). However, when parameters are randomly initialized (LLM-PPO-rand), the LLM fails to comprehend the meaning of the input text and cannot effectively learn from self-play experience.
>
> We further conducted a knowledge probing experiment to test whether LLM can learn game knowledge during self-play. As trained LLMs are not aligned for chatting, we adopt linear probing method [4] for knowledge examination: Specifically, we use trained LLM to extract the representation of the below questions (constructed from the type-effectiveness chart) and then train a logistic regression model on 50% of examples and test on the left examples:
>
> ```
> Question: In Pokémon battles, a {TYPE1} attack is ____ against a {TYPE2} Pokémon?\nA. Super Effective (2x Damage)\nB. Standard (1x Damage)\nC. Not Effective (0.5x Damage)\nD. Zero Effect (0x Damage)\nOutput:
> ```
> Label: $y \in$ {A, B, C, D} (Please see the response to Q2 for how to construct questions)
>
> In below we report the F1 score of each category and weighted F1 before and after self-play:
>
> | Model  | F1 (B)  | F1 (A) | F1 (C) | F1 (D) | F1    | Win Rate|  Battle Score |
> |--------|---------|--------|--------|--------|--------| ------------|------------|
> | Origin | 0.7103  | 0.1200 | 0.2642 | 0.0000 | 0.5231 | 0.1075  | 3.904 |
> | RFT    | 0.7589  | 0.1000 | 0.2593 | 0.0000 | 0.5510 | 0.1161  | 4.194 |
> | DPO    | 0.7477  | 0.1860 | 0.2308 | 0.0000 | 0.5533 | 0.1212 | 4.207 |
>
> We observe that after training-time grounding, LLMs demonstrate improved performance on the type-effectiveness prediction task, showing that LLMs can learn game knowledge from self-play without relying on external sources, positively correlating a higher level of game knowledge with better gameplay performance.

---

> ### Author Response · Authors · 2024-11-20
> **Author Response (II): Response to W2, W3 and questions Q1, Q2**
>
> **W2: Some figure captions are barely readable**
>
> Thank you for your feedback. As you suggested, we have revised the paper to make the captions more readable for readers.
>
> **W3: Appendix B is empty with a title.**
>
> We apologize for this. Appendix B was planned originally for a study on the effect of varying temperatures, which is Table 6 in the main paper.
>
> **Q1: Randomly-matched human players from game server and comments on potential bias if any in the online battles**
>
> The original statement for "random matched human players" in Section 6.2 is "In order to demonstrate higher-level reasoning challenges, we set up online battles with human players **randomly matched** from the game server."
>
> By randomly matched here, we meant that our benchmark triggers the matching on the fly with no conditions attached, and the mechanism for matching in game server is not publicly disclosed (https://pokemonshowdown.com/pages/ladderhelp). However, to respond to the comment by this reviewer, we did further investigation on how human players are matched to one another, and we felt that it is reasonable to assume that the server performs some sort of controlled random-matching by involving factors like the rating and proportion of players across different skill levels.
>
> To address the reviewer's concern, we report the Elo and Glicko scores of the LLM player and its opponent human players, which are indicators of a player's position on the Ladder without potential bias introduced by the matching strategy. The Elo score indicates the relative skill levels of players based on match outcomes, and Glicko is an extension of Elo by incorporating rating reliability (both calculated by the game server).
>
> We report the ratings of LLM players after online battles in the table below and visualize the distribution of ratings and account registration days for matched human players: https://postimg.cc/dZVHQr81
>
> |Metric | Elo | Glicko|
> |:------:|:----:|:------:|
> |Rating | 1156 | 1481|
>
> The game-play capability of a player can be categorized as follows based on the Elo rating:
>
> - **1000-1200:** Elementary level (build up fundamentals)
> - **1200-1400:** Intermediate level (good at hedging guesses and making predictions)
> - **1400-1600:** Advanced level (can get excellent practice for tournament)
> - **1600+:** Master level (top players)
>
> Overall, the most advanced LLM is categorized at the elementary level, indicating substantial room for improvement in the game-play capabilities within our benchmark.
>
> (Reference for game-play capability and Elo rating: https://www.vgcguide.com/what-is-a-good-ladder-rating#:~:text=Battling%20at%201200%20%2D%201400%20ELO,your%20team%20and%20make%20predictions.)
>
> **Q2: The process of generating multiple-choice questions for examining game knowledge and scale of the question dataset**
>
> There are 18 types in Pokémon Battles: Bug, Dark, Dragon, Electric, Fairy, Fighting, Fire, Flying, Ghost, Grass, Ground, Ice, Normal, Poison, Psychic, Rock, Steel, and Water. The type-effectiveness chart (https://pokemondb.net/type, also Figure 6 in Appendix) specifies the damage calculation multipliers for interactions between the 18 attack types and the 18 Pokémon types. Therefore, there are a total of  18*18=324 questions, which can be constructed from the chart (the relationships are asymmetric, e.g., Electric-type attacks deal 2x damage to Water-type Pokémon, whereas Water-type attacks deals 1x damage to Electric-type Pokémon).
>
> An illustrative example:
> ```
> Multi-choice question: In Pokémon battles, a Fire-type attack is ____ against a Grass-type Pokémon.
> A. Super-effective (2x)
> B. Standard (1x)
> C. Ineffective (0.5x)
> D. No effect (0x)
> Answer: A
> ```
>
> Statistics of the dataset:
>
> | Category |# of questions| Ratio |
> |:--------:|:--------------:|:--------:|
> | A        | 51             | 15.74%   |
> | B        | 206            | 63.58%   |
> | C        | 59             | 18.21%   |
> | D        | 8              | 2.47%    |
>
>
> [1] Rafailov, Rafael, et al. "Direct preference optimization: Your language model is secretly a reward model." Advances in Neural Information Processing Systems 36 (2024).
>
> [2] Yuan, Zheng, et al. "Scaling relationship on learning mathematical reasoning with large language models." arXiv preprint arXiv:2308.01825 (2023).
>
> [3] Schulman, John, et al. "Proximal policy optimization algorithms." arXiv preprint arXiv:1707.06347 (2017).
>
> [4] Li, Kenneth, et al. "Inference-time intervention: Eliciting truthful answers from a language model." Advances in Neural Information Processing Systems 36 (2024).
>
> ----
>
> We sincerely thank this reviewer for your insightful comments and suggestions. If the reviewer finds our responses satisfactory, we hope that you would consider raising your scores.

---

> > ### Comment · Reviewer_H5C9 · 2024-11-25
> >
> > Thank you for your response. However, I still have concerns about the contribution of this paper, so I will maintain my score.
> > By the way, I'm not saying that the caption is not readable. I mean the text in some figures (the screenshots) is not clear.

---

> ### Author Response · Authors · 2024-11-26
> **Thanks for your response**
>
> Dear Reviewer H5C9,
>
> Thank you for your feedback. We have slightly enlarged the screenshots to make the text clearer.
>
> Regards,
> Authors

---

### Author Response · Authors · 2024-11-20
**Author summary of initial review**

We sincerely thank all five reviewers for their constructive feedback on our paper submitted to the "datasets and benchmarks" track.

Per the initial review, three reviewers (Reviewer eJmw, Reviewer Tc95, Reviewer mcz9) commented on the innovation of introducing PokéLLMon as a new benchmark for LLM grounding and reasoning research, and four reviewers (Reviewer H5C9, Reviewer eJmw, Reviewer Tc95, and Reviewer mcz9) made special remarks that our experiments were comprehensive and robust.

However, we observe that there are several concerns that we need to carefully address. We list the main concerns and a summary of our responses as follows:

----
**Reviewer H5C9**: Suggest to add a new baseline, and raised concerns on online battle results.

**Author response**: We implemented new reinforcement learning baselines (LLM-PPO and Symbolic-PPO) for training-time grounding. Experiments shows that with a trained value model, learning from self-play becomes more effective, compared to weighing actions based on the win/lose signal of the entire trajectory.

For online battles, we also reported the Elo score and the average account registration age of the opponent players, to objectively measure the gameplay capability in Ladder competitions.

----
**Reviewer eJmw**: Consider adding PPO in the benchmark and code link accessibility.

**Author response**: We added new reinforcement learning baselines (LLM-PPO and Symbolic-PPO) for training-time grounding. See our detailed response for detailed comparison results.

Upon receiving the reviews, we checked why our code link is not accessible, and we learned that the AnonymousGitHub website has set an expiration date for the links, and our PokeLLMon link expired on Oct. 27. We have reset the link: https://anonymous.4open.science/r/PokeLLMon.

----
**Reviewer Loeg**: Discuss the generalizability of enhancing action consistency beyond PokeLLMon, and code link accessibility.

**Author response**: We added new experiments and discussion on the generality of action consistency in other interactive environments, and the efficacy of enhancing consistency in Chain-of-Thoughts in the presence of inconsistency. See our detailed response to Reviewer Loeg.

Upon receiving the reviews, we checked why our code link is not accessible, and we learned that the AnonymousGitHub website has set an expiration date for the links, and our PokeLLMon link expired on Oct. 27. We have reset the link: https://anonymous.4open.science/r/PokeLLMon. We apologize for not knowing the link expiration default from the AnonymousGitHub hosting.

----
**Reviewer Tc95**:  Analyze the failure modes of LLMs and discuss the generality of action consistency.

**Author response**: To better understand the failure of LLM reasoning within PokéLLMon—whether due to inconsistency or the inability to find better strategies—we conducted a study by setting 'consecutive switching' as inadmissible to disentangle the effect of reasoning in finding better strategies. Experiments show that under this controlled setting, LLMs indeed gain benefits from reasoning, suggesting the failure mainly comes from the inconsistency, and the efficacy of LastThoughts stems not only from mitigating inconsistency but also from enhanced reasoning.

We discussed the generality of action consistency with explanations, and experimental results from another environment to demonstrate the efficacy of enhancing consistency.

----

**Reviewer mcz9**: Provide sanity check, knowledge examination study and an inadmissible action example

**Author response**: For sanity check, we set “consecutive switching” as inadmissible to disentangle the effect of reasoning in finding better strategies. Experiments show that CoT indeed outperforms IOPrompt in this setting. We presented additional experiments on knowledge examination study to compare results of before and after the self-play, which shows that LLMs do acquire knowledge from interactions. Finally, we presented an example to illustrate the inadmissible action of open-source LLMs.

We thank the reviewers for the helpful comments and we provide a more detailed response in individual comments for each reviewer. We look forward to further discussions during the rebuttal period.

---

### Comment · Area_Chair_wiwx · 2024-11-25

Dear Reviewers,


This is a friendly reminder that the discussion will end on Nov. 26th (anywhere on Earth). If you have not already, please take a close look at all reviews and author responses, and comment on whether your original rating stands.


Thanks,

AC

---

> ### Comment · Area_Chair_wiwx · 2024-11-29
>
> This is a friendly reminder that the last day that reviewers can post a message to the authors is Dec. 2nd (anywhere on Earth). If you have not already, please take a close look at all reviews and author responses, and comment on whether your original rating stands.
>
> Thanks,
>
> AC

---

### Note · Authors · 2025-01-08

I have read and agree with the venue's withdrawal policy on behalf of myself and my co-authors.